# A single-cell atlas of spatial and temporal gene expression in the mouse cranial neural plate

Eric R Brooks[1,2], Andrew R Moorman[3], Bhaswati Bhattacharya[2], Ian S Prudhomme[2], Max Land[3], Heather L Alcorn[2], Roshan Sharma[3], Dana Pe'er[3], Jennifer A Zallen[2]*

[1]Department of Molecular Biomedical Sciences, College of Veterinary Medicine, North Carolina State University, Raleigh, United States; [2]Howard Hughes Medical Institute and Developmental Biology Program, Sloan Kettering Institute, New York, United States; [3]Howard Hughes Medical Institute and Computational and Systems Biology Program, Sloan Kettering Institute, New York, United States

## eLife Assessment

This comprehensive scRNAseq atlas of the cranial region during neural induction, patterning, and morphogenesis provides a **fundamental** demonstration of how different cell fates are organized in specific spatial patterns along the anterior-posterior and medial-lateral axes within the developing neural tissue. The **compelling** data are analyzed with a rigorous computational approach, and the data revealed both known and novel genes differentially expressed along rostro-caudal and medio-lateral axes. This will be a helpful resource for researchers studying brain development.

*For correspondence:
zallenj@mskcc.org

**Abstract** The formation of the mammalian brain requires regionalization and morphogenesis of the cranial neural plate, which transforms from an epithelial sheet into a closed tube that provides the structural foundation for neural patterning and circuit formation. Sonic hedgehog (SHH) signaling is important for cranial neural plate patterning and closure, but the transcriptional changes that give rise to the spatially regulated cell fates and behaviors that build the cranial neural tube have not been systematically analyzed. Here, we used single-cell RNA sequencing to generate an atlas of gene expression at six consecutive stages of cranial neural tube closure in the mouse embryo. Ordering transcriptional profiles relative to the major axes of gene expression predicted spatially regulated expression of 870 genes along the anterior-posterior and mediolateral axes of the cranial neural plate and reproduced known expression patterns with over 85% accuracy. Single-cell RNA sequencing of embryos with activated SHH signaling revealed distinct SHH-regulated transcriptional programs in the developing forebrain, midbrain, and hindbrain, suggesting a complex interplay between anterior-posterior and mediolateral patterning systems. These results define a spatiotemporally resolved map of gene expression during cranial neural tube closure and provide a resource for investigating the transcriptional events that drive early mammalian brain development.

## Introduction

Development of the mammalian brain requires coordination between the genetic programs that generate cell fate and the dynamic and spatially regulated cell behaviors that establish tissue structure. The cranial neural plate is an epithelial sheet that undergoes neuronal differentiation and structural remodeling to produce functionally distinct brain regions in the forebrain, midbrain, and hindbrain.

**eLife digest** Brain development in mammals starts when a sheet of cells rolls into a tube, a process called neural tube closure. Missteps in this process are among the most common structural brain malformations in humans, affecting about one in 2,000 births. Learning more about this dynamic process will help scientists understand why these common neural tube defects occur and elucidate the mysteries of early brain development.

Neural tube closure involves dramatic changes in tissue architecture that require carefully orchestrated changes in cell behavior and gene expression. Scientists have identified many key signals that set these processes in motion. However, an important missing piece was a comprehensive description of the gene expression changes that occur during closure, which provide a blueprint for the developing brain.

Using single-cell RNA sequencing, Brooks et al. assembled a two-dimensional map of the expression of thousands of genes during neural tube closure in the mouse embryo. Starting with a few well-known genes with localized expression patterns, they were able to predict the expression patterns of 870 genes during the process. This map matched known gene expression with high accuracy and predicted expression patterns for hundreds of previously uncharacterized genes. Moreover, they showed that activating a key developmental pathway, known as Sonic hedgehog signaling, dramatically reshaped gene expression along the width and length of the neural tube.

Brooks et al. provide a two-dimensional map of gene expression in the mouse cranial neural plate that will help scientists trace the earliest steps in brain development. This work provides a road map to understanding the processes that generate the remarkable organization and functions of the mammalian brain.

The forebrain gives rise to structures including the cerebral cortex, thalamus, and hypothalamus, the midbrain generates the visual and auditory systems, and the hindbrain forms the cerebellum and other structures that control cognitive, emotional, autonomic, and motor functions (*Moore et al., 2014*; *Chen et al., 2017*). The generation of these three domains is controlled by a cartesian landscape of transcriptional information that directs cell fate and behavior along the anterior-posterior and mediolateral axes of the cranial neural plate, guided by conserved signals in the WNT, BMP, SHH, and FGF families and retinoic acid (*Liu and Joyner, 2001a*; *Wurst and Bally-Cuif, 2001*; *Maden, 2007*; *Kicheva and Briscoe, 2023*). However, the transcriptional programs that drive neural tube patterning and closure to provide a blueprint for the fine-scale organization of the mammalian brain are not well understood.

Single-cell RNA sequencing (scRNA-seq) is a powerful tool for analyzing gene expression during development (*Tam and Ho, 2020*). This approach has been used to describe gene expression in several mouse organs (*Cardoso-Moreira et al., 2019*; *Delile et al., 2019*; *Nowotschin et al., 2019*; *Soldatov et al., 2019*; *de Soysa et al., 2019*; *La Manno et al., 2021*; *Zalc et al., 2021*) and entire mouse embryos at different stages (*Argelaguet et al., 2019*; *Cao et al., 2019*; *Cheng et al., 2019*; *Pijuan-Sala et al., 2019*; *Mittnenzweig et al., 2021*; *Qiu et al., 2022*; *Sampath Kumar et al., 2023*; *Qiu et al., 2024*). Region-specific transcriptional signatures in the mouse cranial neural plate are first detected between embryonic days 7.0 and 8.5 (*Pijuan-Sala et al., 2019*; *La Manno et al., 2021*; *Lohoff et al., 2022*; *Qiu et al., 2022*; *Sampath Kumar et al., 2023*) and gene expression profiles in neuronal populations have been characterized in detail at later stages of brain development (*Rosenberg et al., 2018*; *Li et al., 2019*; *Wizeman et al., 2019*; *Wang et al., 2020*; *Di Bella et al., 2021*; *La Manno et al., 2021*; *Ruan et al., 2021*; *Langlieb et al., 2023*; *Yao et al., 2023*; *Zhang et al., 2023*; *Qiu et al., 2024*). However, although the transcriptional changes that partition cranial neuroepithelial progenitors into distinct domains are beginning to be identified, significant questions remain. First, cranial neural plate differentiation and closure are accompanied by dramatic changes in gene expression, and the transcriptional dynamics that give rise to the metameric organization of the forebrain, midbrain, and hindbrain over time remain opaque. Second, the patterned transcriptional programs that delineate distinct structural and functional domains within these regions, and how they are spatially organized along the anterior-posterior and mediolateral axes, have not been systematically analyzed. Third, how gene expression profiles are regulated by morphogen patterning systems, and

the transcriptional basis of pathological signaling outcomes such as the effects of increased Sonic hedgehog signaling on cranial neural tube patterning and closure (*Murdoch and Copp, 2010*; *Brooks et al., 2020*), are not well understood.

To address these questions, we used scRNA-seq to analyze the spatial and temporal regulation of gene expression in the developing mouse cranial neural plate. We obtained gene expression profiles for 39,463 cells from the cranial region of mouse embryos at six stages from day 7.5 to day 9.0 of embryonic development, during which the cranial neuroepithelium undergoes dynamic changes in tissue patterning and organization, culminating in neural tube closure. Analysis of the 17,695 cranial neural plate cells in this dataset revealed distinct gene expression trajectories over time in the developing forebrain, midbrain, and hindbrain. In addition, we used this dataset to computationally reconstruct a high-resolution map of gene expression in the E8.5–9.0 cranial neural plate at late stages of neural tube closure. This map predicted spatially regulated expression for 870 genes along the anterior-posterior and mediolateral axes of the cranial neural plate, including 687 genes whose expression had not previously been characterized, and recapitulated the expression patterns of known genes with over 85% accuracy. Finally, we used scRNA-seq analysis to systematically investigate the consequences of activated SHH signaling, which disrupts neural tube patterning and closure. This analysis revealed region-specific transcriptional responses to SHH signaling in the forebrain, midbrain, and hindbrain, suggesting complex interactions between anterior-posterior and mediolateral patterning systems. Together, these results define a spatially and temporally resolved map of gene expression during cranial neural tube closure and provide a resource for examining the early transcriptional events that drive mammalian brain development.

## Results

### Temporal evolution of region-specific transcriptional programs in the cranial neural plate

To characterize gene expression during cranial neural tube patterning and closure, we performed scRNA-seq analysis of manually dissected cranial regions from mouse embryos at six consecutive stages spanning embryonic days (E) 7.5–9.0 of development (Materials and methods). This dataset includes two stages before neural tube closure (0 somites and 1–2 somites, corresponding to E7.5–7.75), two stages during early closure as the neural plate bends and elevates (3 somites and 4–6 somites, E8.0–8.25), and two stages during late closure as the lateral borders meet to form a closed tube (7–9 somites and 10+somites, E8.5–9.0) (*Figure 1A and B*). Using stringent criteria to filter out doublets and low-quality cells (Materials and methods), we obtained 39,463 cranial cells from 30 samples (3–5 embryos/replicate, 4–6 replicates/stage), with a median library depth of >42,000 transcripts (unique molecular identifiers or UMIs)/cell and >5900 genes/cell (average 1.56 reads/UMI; *Supplementary file 1*). Using the PhenoGraph clustering method (*Levine et al., 2015*) after correcting for cell-cycle stage (Materials and methods), we identified 29 transcriptionally distinct clusters in the cranial region that represent 7 cell types based on known markers: neural plate (17,695 cells), mesoderm (13,952 cells), neural crest (2993 cells), non-neural ectoderm (2252 cells), endoderm (2062 cells), notochord (449 cells), and blood cells (60 cells; *Figure 1C*, *Figure 1—figure supplement 1*, *Supplementary file 1* and *Supplementary file 2*). Thus, this dataset captures all major cell populations in the cranial region of the mouse embryo.

To examine gene expression in the cranial neural plate in more detail, we reclustered the 17,695 cranial neural plate cells in our dataset and identified 15 clusters using PhenoGraph analysis (*Figure 1— figure supplement 2A, B*, *Supplementary file 3*). These clusters did not feature markers of neuronal or radial glial cells, consistent with the later onset of neurogenesis in the cranial neural plate (*Götz and Huttner, 2005*), although we observed modest expression of neuron-specific cytoskeletal regulators (*Figure 1—figure supplement 2C*). Clusters instead appeared to distinguish spatial and temporal properties of cells, as transcriptional profiles were strongly correlated with embryonic stage for all cranial populations (*Figure 1D*) and within the cranial neural plate (*Figure 1E*), and gene expression in the cranial neural plate was strongly correlated with anterior-posterior location (*Figure 1F–J*).

To disambiguate spatial and temporal features of transcriptional regulation, we used region-specific markers to computationally divide the cranial neural plate into four populations: the developing forebrain (*Otx2, Six3*; 6060 cells), the developing midbrain and rhombomere 1 (midbrain/r1, encompassing

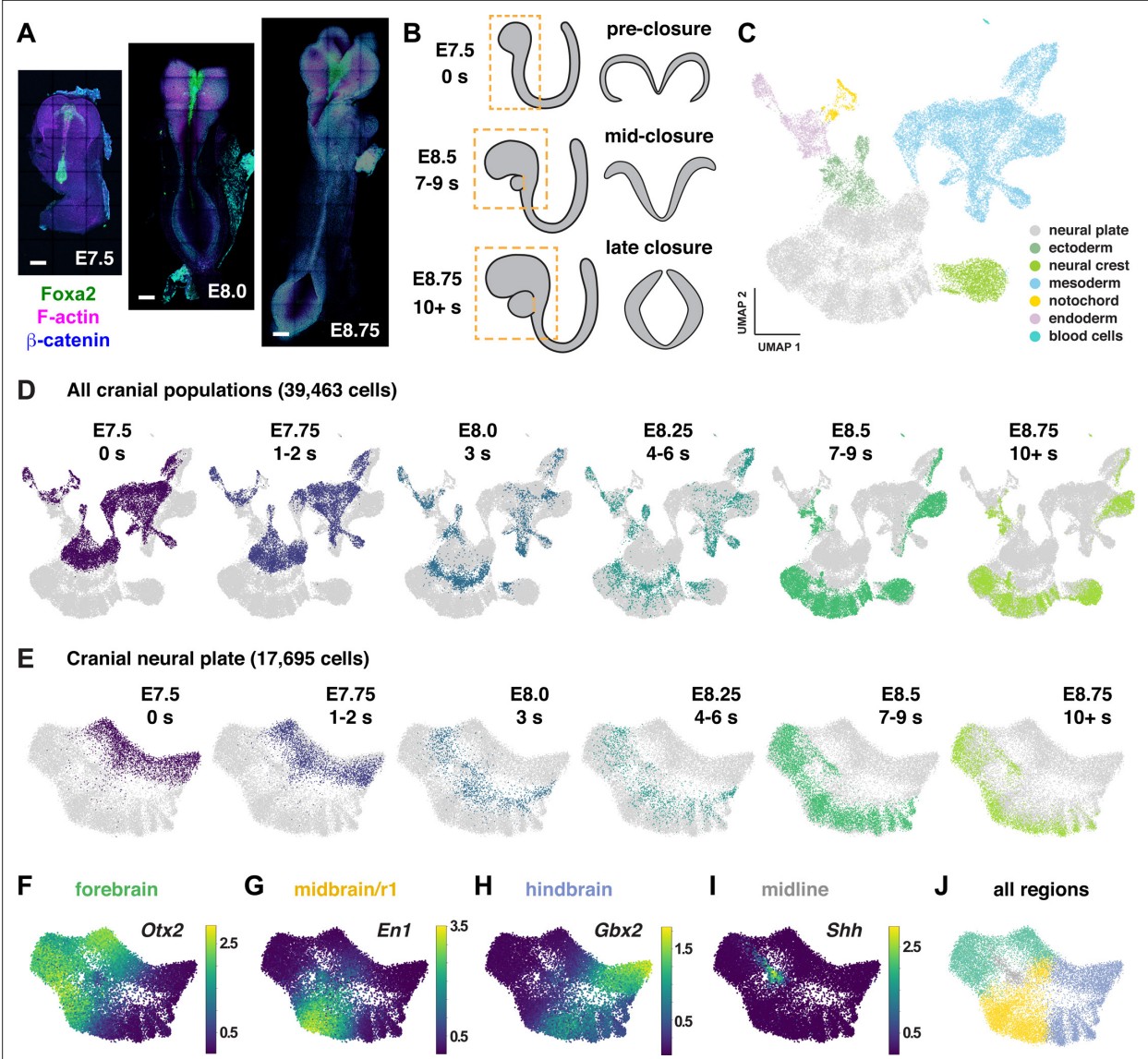

**Figure 1.** Construction of a single-cell RNA expression atlas during mouse cranial neural tube closure. (**A**) Images of mouse embryos spanning the stages of cranial neural tube closure. Bars, 100 µm. (**B**) Schematics of cranial tissues collected for scRNA-sequencing (dashed boxes, left) and neural tube closure in the midbrain (right) at the indicated stages. The heart was removed at later stages. (**C, D**) UMAP projections of all cranial cell populations analyzed colored by cell type (**C**) or embryonic stage (**D**). (**E**) UMAP projections of cranial neural plate cells colored by embryonic stage. (**F–I**) UMAP projections of cranial neural plate cells colored by normalized expression of genes primarily associated with the forebrain (*Otx2*), midbrain and rhombomere 1 (*En1*), hindbrain (*Gbx2*), and ventral midline (*Shh*). (**J**) UMAP projection of cranial neural plate cells colored by neural plate region.

The online version of this article includes the following figure supplement(s) for figure 1:

**Figure supplement 1.** Assignment of cell identities in the mouse cranial region.

**Figure supplement 2.** Assignment of cell identities in the mouse cranial neural plate.

the midbrain-hindbrain boundary; *En1, En2, Fgf8, Pax2, Pax5, Pax8, Wnt1*; 5306 cells), presumptive rhombomeres 2–5 (referred to as the hindbrain; *Egr2, Gbx1, Gbx2, Hoxa2, Hoxb2, Hoxb3*; 5568 cells), and the ventral midline (*Foxa2, Nkx6-1, Ptch1, Shh*), also known as the floor plate (761 cells). We then applied diffusion component analysis to identify gene expression trends in each population (Materials and methods). The top diffusion component (DC0) in each region correlated with embryo stage, suggesting that DC0 can be used to analyze gene expression over time (***Figure 2A–C***, ***Figure 2— figure supplement 1A–F***). Consistent with the use of DC0 to represent developmental time, E-cad-herin (*Cdh1*) transcript levels decreased and N-cadherin (*Cdh2*) transcript levels increased relative

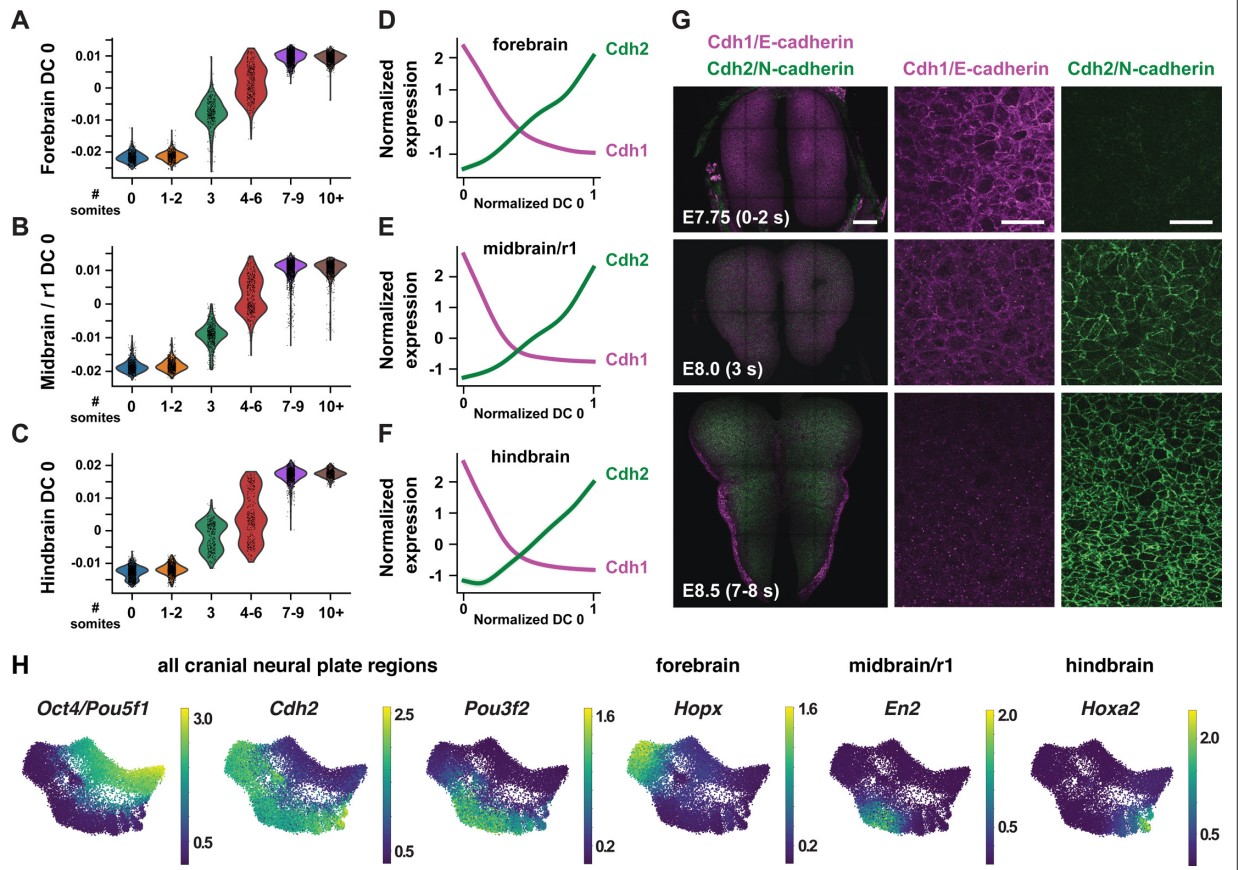

**Figure 2.** Temporal changes in gene expression reveal shared and region-specific transcriptional trajectories. (**A–C**) Cells from the forebrain (**A**), midbrain/r1 (**B**), and hindbrain (**C**) at progressive stages of neural plate development, plotted relative to the time-correlated diffusion component 0 (DC0) in each region. (**D–F**) Normalized expression of *E-cadherin* (*Cdh1*) and *N-cadherin* (*Cdh2*) plotted relative to the normalized time-correlated diffusion component (DC0) in each region. (**G**) E-cadherin protein (magenta) is lost from cell-cell junctions and N-cadherin protein (green) accumulates at cell-cell junctions between E7.75 and E8.5 in the mouse cranial neural plate. (**H**) UMAP projections of cranial neural plate cells colored by normalized gene expression, showing examples of genes that are downregulated or upregulated throughout the cranial neural plate (left) and genes that are specifically upregulated in the forebrain, midbrain/r1, or hindbrain. Bars, 100 μm (left panels in G), 20 μm (middle and right panels in G).

The online version of this article includes the following figure supplement(s) for figure 2:

**Figure supplement 1.** Analysis of gene expression trends in the developing forebrain, midbrain/r1, and hindbrain.

**Figure supplement 2.** Examples of genes that are temporally regulated throughout the cranial neural plate.

**Figure supplement 3.** Examples of genes that are upregulated in the forebrain, midbrain/r1, or hindbrain.

to DC0 in each region (*Figure 2D–F*), correlating with stage-specific changes in the corresponding proteins (*Figure 2G*). These results show that DC0 captures the known temporal shift in cadherin expression in the cranial neural plate (*Kimura et al., 1995*; *Radice et al., 1997*; *Lee et al., 2007*) and demonstrate that this transition is associated with a global change in transcriptional state.

As DC0 correlates with developmental stage, we investigated gene expression dynamics by aligning the expression of each gene in the forebrain, midbrain/r1, and hindbrain relative to DC0 for each region and clustering genes based on their expression along this axis (*Figure 2—figure supplement 1G–I*, *Supplementary file 4*; Materials and methods). A number of genes were downregulated relative to DC0 throughout the cranial neural plate, including pluripotency factors (*Oct4/Pou5f1*), epithelial identity genes (*Cdh1, Epcam*), mitochondrial regulators (*Chchd10, Eci1*), epigenetic factors (*Dnmt3b*), and the Alzheimer's-associated gene *Apoe* (*Figure 2H*, *Figure 2—figure supplement 2A*). In addition, several genes were upregulated relative to DC0 throughout the cranial neural plate, including neurogenic factors (*Ndn, Pou3f2, Sox21*), cell division regulators (*Mycl, Mycn*), and genes involved in neural tube closure (*Cdh2, Palld*; *Figure 2H*, *Figure 2—figure supplement 2B*). These

results indicate that cells in the E7.5-E9.0 cranial neural plate transition from a pluripotent to a pro-neurogenic and morphogenetically active state.

In addition to changes in gene expression throughout the cranial neural plate, several genes were specifically upregulated in the forebrain, midbrain/r1, hindbrain, or multiple brain regions (*Figure 2H*, *Figure 2—figure supplement 2C*, *Figure 2—figure supplement 3*, *Supplementary file 4*). These include genes encoding specific transcriptional regulators in the forebrain (*Arx, Barhl2, Emx2, Foxd1, Foxg1, Hopx, Lhx2, Six6*), midbrain/r1 (*En2, Pax8*), and hindbrain (*Egr2, Hoxa2, Msx3*), as well as transcriptional regulators expressed in multiple regions (*Irx2, Otx1, Pax6*). In addition, several genes involved in cell-cell communication were upregulated in a region-specific fashion, including *Dkk1, Scube1, Scube2,* and *Wnt8b* in the forebrain, *Cldn10, Fgf17*, and *Spry1* in the midbrain/r1 region, and *Fgf3, Plxna2, Robo2,* and *Sema4f* in the hindbrain. These results are consistent with the emergence of region-specific cell identities along the anterior-posterior axis of the E7.5-E9.0 cranial neural plate (*Liu and Joyner, 2001a*; *Wurst and Bally-Cuif, 2001*; *Lohoff et al., 2022*), and provide an opportunity to systematically define the transcriptional basis of these developmental patterning events.

## Modular organization of gene expression along the anterior-posterior axis of the cranial neural plate

The emergence of distinct gene expression signatures in the presumptive forebrain, midbrain/r1, and hindbrain suggests that this dataset can be used to identify factors that contribute to distinct cell fates and behaviors along the anterior-posterior axis. To characterize the spatial organization of transcription in the cranial neural plate, we pooled cranial neural plate cells from E8.5 (7–9 somites) and E8.75–9.0 (10+somites) embryos, when strong regional patterns of gene expression were apparent, and we reapplied diffusion component analysis on the aggregated data (Materials and methods). The top diffusion component in the pooled E8.5–9.0 dataset (DC0) reproduced the anterior-posterior order of known markers of the forebrain (*Six3, Otx1, Otx2*), midbrain/r1 (*En1, Wnt1*), and hindbrain (*Gbx2*) regions, indicating that this diffusion component correctly orders cells relative to the anterior-posterior axis (*Figure 3A–C*, *Figure 3—figure supplement 1*, *Supplementary file 5*). In addition, DC0 recapitulated the anterior-posterior order of markers of more refined transcriptional domains, such as the future telencephalon (*Foxg1, Lhx2*), diencephalon (*Barhl2*), midbrain-hindbrain boundary (*Fgf8*), and rhombomeres 2–5 (*Hoxb1, Hoxb2, Egr2*; *Figure 3B and C*). The second diffusion component in the pooled E8.5–9.0 dataset (DC1) also displayed a strong anterior-posterior signature, with DC1 separating genes expressed in the midbrain/r1 domain from genes expressed in the forebrain and hindbrain (*Figure 3—figure supplement 1*, *Supplementary file 5*). These results indicate that DC0 and DC1 correlate with anterior-posterior pattern, with DC0 ordering cells along the anterior-posterior axis.

To identify genes with spatially regulated transcriptional signatures along the anterior-posterior axis, we aligned all genes along DC0 and clustered them based on their dynamics along this axis (Materials and methods). Further, we used HotSpot (*DeTomaso et al., 2019*), a method to identify genes whose expression reflects statistically significant local similarity among cells (including spatial proximity), to distinguish genes with significant expression along DC0. This analysis identified 483 genes in 11 clusters that are predicted to be patterned relative to the anterior-posterior axis of the cranial neural plate (*Figure 3D*, *Figure 3—figure supplement 2*, *Supplementary file 6*). By comparing with markers for specific anterior-posterior domains (*Figure 3C*), we assigned these clusters to the future forebrain (clusters 0 and 7), midbrain/r1 (clusters 3 and 5), and hindbrain (clusters 4, 6, and 10), with some clusters spanning multiple domains (clusters 1, 2, 8, and 9; *Figure 3D*). We note that although the midbrain and rhombomere 1 shared clear transcriptional signatures that distinguish them from other regions of the cranial neural plate, there were also clear differences between these domains, consistent with their respective midbrain and hindbrain identities. Together, these results provide a comprehensive description of the modular organization of gene expression in the cranial neural plate and reveal that hundreds of genes are expressed in a small number of discrete domains along the anterior-posterior axis, corresponding to the metameric organization of the forebrain, midbrain/r1, and hindbrain regions.

To assess the accuracy of the predicted expression patterns, we compared the predictions with published gene expression data in the Mouse Genome Informatics Gene Expression Database (http://informatics.jax.org; *Baldarelli et al., 2021*). Of 483 genes predicted to be patterned along

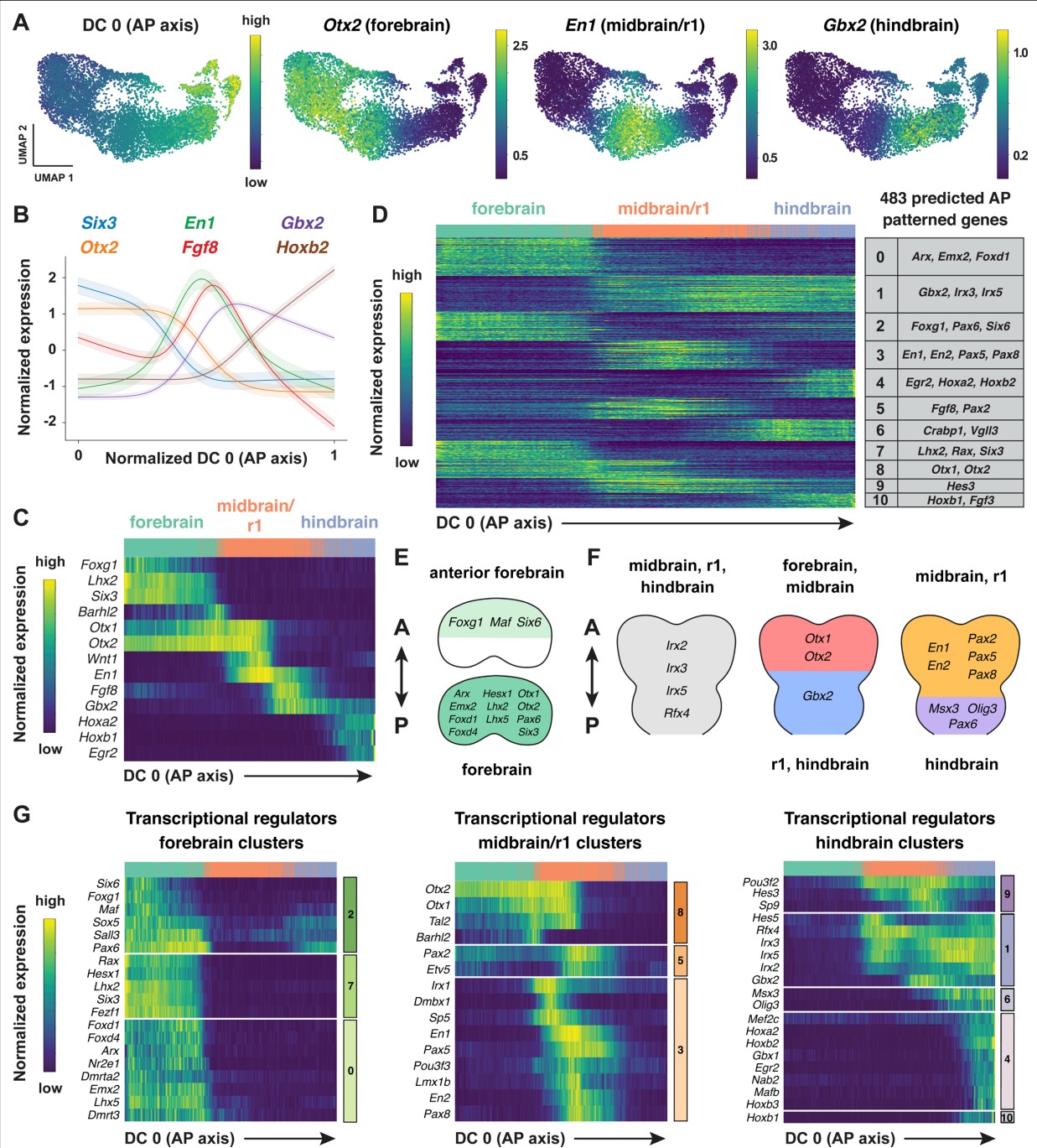

**Figure 3.** Region-specific patterns of gene expression along the anterior-posterior axis. (**A**) UMAP projections of cranial neural plate cells from E8.5–9.0 embryos colored by their value along the anterior-posterior-correlated diffusion component (DC0) (left) or by the normalized expression of markers primarily associated with the forebrain (*Otx2*), midbrain/r1 (*En1*), and hindbrain (*Gbx2*). (**B, C**) Line plot (**B**) and heatmap (**C**) showing the normalized expression of known markers of anterior-posterior identity relative to DC0. DC0 correctly orders forebrain, midbrain/r1, and hindbrain markers relative to their known positions along the anterior-posterior axis. (**D**) Heatmap showing the normalized expression of 483 genes with high information content relative to DC0. Gene expression profiles are grouped into 11 clusters based on similarities in expression along DC0. Examples of genes in each cluster are listed on the right. (**E, F**) Schematics showing the predicted expression of example transcriptional regulators along the anterior-posterior axis. Anterior (A), posterior (P). (**G**) Heatmaps showing the normalized expression of example transcriptional regulators relative to DC0. Heatmaps show one gene per row, one cell per column, with the cells in each row ordered by their value along DC0. Colored bars (right) show the cluster identity relative to DC0. Cells assigned to the forebrain, midbrain/r1, or hindbrain are indicated at the top of each heatmap.

The online version of this article includes the following figure supplement(s) for figure 3:

*Figure 3 continued on next page*

*Figure 3 continued*

**Figure supplement 1.** Diffusion component analysis of E8.5–9.0 neural plate cells.

**Figure supplement 2.** Clustering analysis reveals distinct patterns of gene expression along DC0 in the E8.5–9.0 cranial neural plate.

the anterior-posterior axis, 162 had published images of in situ hybridization or protein immuno-staining experiments in the E8.5–9.0 cranial region (Theiler stages 12–14; *Theiler, 1989*). We found that 141 of the 162 genes for which data were available (87%) were expressed in region-specific domains along the anterior-posterior axis that were consistent with the patterns predicted by our analysis (*Supplementary file 7*, Materials and methods). The remaining 21 genes were expressed in the predicted domain(s) in addition to one or more regions not predicted by our analysis, likely due to the increased sensitivity of in situ hybridization compared with scRNA-seq (*Lohoff et al., 2022*). Therefore, ordering gene expression relative to DC0 recapitulated the expression patterns of 141 known genes and predicted patterned expression along the anterior-posterior axis for an additional 321 genes whose expression in the developing cranial neural plate has not previously been analyzed.

To identify region-specific transcriptional regulators that could contribute to these spatial patterns, we compared the list of genes with high information content relative to DC0 with transcriptional regulators predicted by the KEGG BRITE and Gene Ontology databases (*Ashburner et al., 2000*; *Aleksander et al., 2023*; *Kanehisa et al., 2023*; Materials and methods). This approach identified 123 genes encoding transcription factors that are predicted to be spatially regulated along the anterior-posterior axis of the E8.5–9.0 cranial neural plate (54 shown in *Figure 3E–G*; see *Supplementary file 8* for full list). These include *Otx2*, a homeodomain transcription factor that is required to form the forebrain and midbrain (*Acampora et al., 1995*; *Matsuo et al., 1995*; *Ang et al., 1996*), as well as genes required for specific aspects of development in the forebrain (*Foxg1, Hesx1, Rax, Six3*; *Xuan et al., 1995*; *Mathers et al., 1997*; *Dattani et al., 1998*; *Lagutin et al., 2003*), midbrain, or rhombomere 1 (*En1, En2, Fgf8, Gbx2, Pax2, Pax5*; *Joyner et al., 1991*; *Wurst et al., 1994*; *Crossley et al., 1996*; *Favor et al., 1996*; *Torres et al., 1996*; *Wassarman et al., 1997*; *Millet et al., 1999*; *Schwarz et al., 1999*; *Liu and Joyner, 2001b*; *Li et al., 2002*; *Chi et al., 2003*), Together, these results indicate that this dataset links known regulators of cell fate and behavior to defined anterior-posterior expression domains in the cranial neural plate and predicts spatially regulated expression for hundreds of previously uncharacterized genes.

## Patterned gene expression along the mediolateral axis of the midbrain and rhombomere 1

In addition to patterning along the anterior-posterior axis, cell fate determination along the mediolateral axis of the cranial neural plate is essential for neural tube closure, neuronal differentiation, and neural circuit formation (*Shimamura et al., 1997*; *Agarwala et al., 2001*; *Hoch et al., 2009*). However, although spatially regulated gene expression along the dorsal-ventral axis has been systematically characterized in the developing spinal cord (*Delile et al., 2019*; *Sampath Kumar et al., 2023*), the transcriptional cascades that govern mediolateral patterning in the cranial neural plate are not well understood. To elucidate the molecular basis of this mediolateral patterning, we examined whether other diffusion components correlate with mediolaterally patterned gene expression in the E8.5–9.0 cranial neural plate. In contrast to DC0 and DC1, which correlate with markers of anterior-posterior pattern, DC2 was positively correlated with laterally expressed genes (*Pax3, Pax7, Tfap2a, Zic1*) and negatively correlated with markers of the ventral midline (floor plate; *Foxa2, Nkx6-1, Ptch1, Shh*; *Figure 4A–C*, *Supplementary file 5*). These results indicate that DC2 can be used to order genes with respect to the mediolateral axis.

To identify genes with mediolaterally patterned expression in the cranial neural plate, we aligned all genes expressed in the E8.5–9.0 cranial neural plate along DC2 and clustered them based on their expression along this axis (Materials and methods). This analysis identified 253 genes in 7 clusters that display high information content along DC2 and are predicted to be mediolaterally patterned (*Figure 4D*, *Figure 4—figure supplement 1A, B*, *Supplementary file 6*). These include four clusters (0, 4, 5, and 6) predicted to have increased expression in the midline and two clusters (1 and 2) predicted to have increased expression in lateral domains. As genes in clusters 1 and 2 tended to be expressed in the midbrain/r1 region based on the UMAP representations, this suggested that DC2

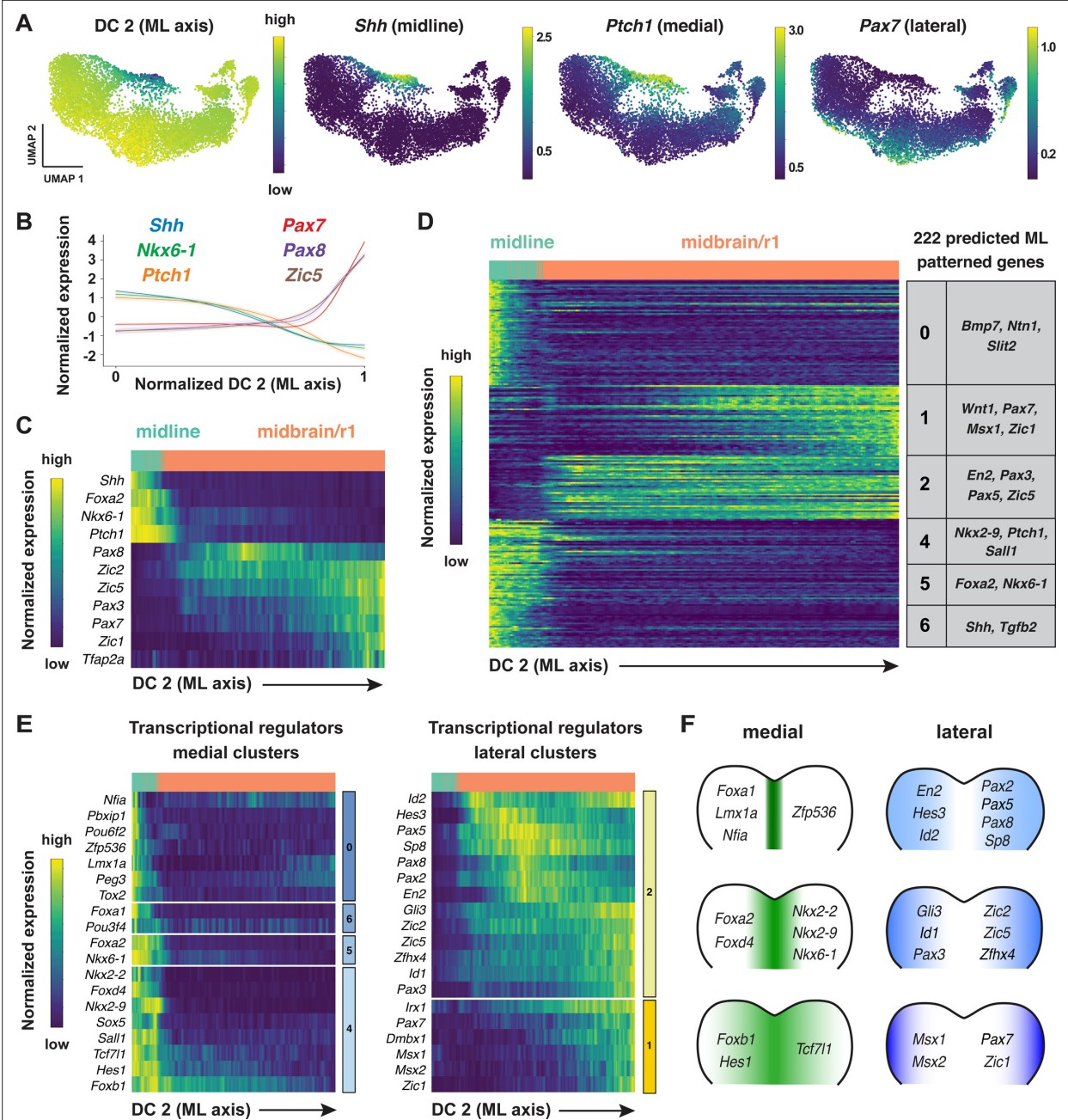

**Figure 4.** Patterned gene expression along the mediolateral axis of the developing midbrain and rhombomere 1. (**A**) UMAP projections of cranial neural plate cells from E8.5–9.0 embryos colored by their value along the mediolaterally correlated diffusion component (DC2) (left) or by the normalized expression of markers for the ventral midline (*Shh*), medial cells (*Ptch1*), or lateral cells (*Pax7*). (**B, C**) Line plot (**B**) and heatmap (**C**) showing the normalized expression of known markers of mediolateral cell identity relative to DC2 in the midbrain/r1. DC2 correctly orders midline, medial, and lateral markers relative to their known positions along the mediolateral axis. (**D**) Heatmap showing the normalized expression of 222 genes with high information content relative to DC2. Gene expression profiles are grouped into seven clusters based on similarities in expression along DC2. Examples of genes in each cluster are listed on the right. Cluster 3 (not shown) displayed divergent UMAP patterns associated with anterior-posterior patterning and was excluded from further analysis. (**E**) Heatmaps showing the normalized expression of specific transcriptional regulators relative to DC2. (**F**) Schematics showing the predicted expression of example transcriptional regulators along the mediolateral axis of the midbrain/r1. Heatmaps show one gene per row, one cell per column, with the cells in each row ordered by their value along DC0. Colored bars (right) show the cluster identity relative to DC2. Cells assigned to the midline or midbrain/r1 are indicated at the top of each heatmap.

The online version of this article includes the following figure supplement(s) for figure 4:

**Figure supplement 1.** Clustering analysis reveals distinct patterns of gene expression along DC2 in the E8.5–9.0 cranial neural plate.

primarily captures the mediolateral pattern in this region. We therefore excluded cluster 3, which contains genes expressed largely outside of this region, from further analysis and focused our analysis of DC2 on the midbrain/r1 and midline cell populations.

To validate the predicted expression patterns, we compared gene expression along DC2 with expression data in the Mouse Genome Informatics Gene Expression Database. Of 222 genes predicted to be patterned along DC2 (excluding cluster 3), 65 genes had published expression data at these stages and 56 (86%) were consistent with the patterns predicted by our analysis (*Supplementary file 7*, Materials and methods). Thus, ordering gene expression relative to DC2 reproduced the expression of 56 known genes and predicted mediolaterally patterned expression for an additional 157 genes whose expression has not previously been characterized in the cranial neural plate.

Comparison of genes patterned along DC2 with known transcriptional regulators identified 78 transcriptional regulators predicted to be patterned along the mediolateral axis of the developing midbrain/r1 region (38 shown in *Figure 4E and F*; see *Supplementary file 8* for full list). Genes predicted to be expressed in medial clusters include effectors of SHH signaling (*Foxa2, Nkx6-1, Nkx2-2*, and *Nkx2-9*; *Sasaki et al., 1997*; *Briscoe et al., 1999*; *Sander et al., 2000*; *Dessaud et al., 2008*; *Ribes and Briscoe, 2009*; *Nishi et al., 2015*). In addition, genes predicted to be expressed in lateral clusters include factors that have been shown to promote dorsal cell fates (*Msx1, Msx2*; *Bach et al., 2003*), suppress neuronal differentiation (*Hes1, Hes3*; *Hirata et al., 2001*; *Hatakeyama et al., 2004*), and regulate cranial neural tube closure (*Gli3, Sp8, Zic2, Zic5*; *Nagai et al., 2000*; *Bell et al., 2003*; *Treichel et al., 2003*; *Inoue et al., 2004*). A subset of genes was expressed in continuously increasing or decreasing patterns relative to DC2, reminiscent of gradients (*Figure 4—figure supplement 1C–F*). Together, these results indicate that this dataset can be used to predict spatially regulated gene expression patterns along the mediolateral axis of the developing midbrain and rhombomere 1.

## An integrated framework for analyzing cell identity in multiscale space

Patterning and morphogenesis of the cranial neural plate require the integration of positional information along the anterior-posterior and mediolateral axes. Nearly one-third of genes that were clustered relative to DC0, and nearly two-thirds of genes that were clustered relative to DC2, were also patterned along the orthogonal axis (*Supplementary file 6*), suggesting a high degree of overlap between anterior-posterior and mediolateral patterning systems. We therefore sought to develop a comprehensive two-dimensional framework to capture both anterior-posterior and mediolateral information. To analyze the full cartesian landscape of gene expression, we extended our approach to all high-eigenvalue diffusion components in the E8.5–9.0 cranial neural plate (*Figure 3—figure supplement 1*), after confirming that each correlated with subsets of known spatial markers in the neural plate (Materials and methods). This analysis identified 870 genes with high mutual information content with respect to anterior-posterior and mediolaterally correlated diffusion components (*Figure 5A and B*). Moreover, because we used the full diffusion space, the resulting clusters integrate more than one cartesian axis at once and capture more fine-grained spatial patterns. These include the genes identified in our analyses of DC0 (anterior-posterior pattern in *Figure 3*) and DC2 (mediolateral pattern in *Figure 4*), as well as additional genes that were not identified from analysis of a single diffusion component (*Figure 5B*, *Supplementary file 6*), suggesting that these expression patterns reflect inputs from a combination of anterior-posterior and mediolateral spatial cues.

To capture both coarse- and fine-grained aspects of gene expression patterns in two dimensions, we clustered genes at different levels of stringency in their pairwise correlations (Materials and methods), referred to here as distance cutoffs (*Figure 5C*, *Supplementary file 6*). At the largest distance cutoff (lowest clustering stringency, D=10), 870 genes were assigned to 15 clusters that are predicted to share broad features of anterior-posterior and mediolateral identity (*Figure 5—figure supplement 1*, *Supplementary file 6*). At an intermediate distance cutoff (D=6), individual clusters corresponded to more refined spatial domains, such as the early telencephalon (*Foxg1, Lhx2*), diencephalon (*Barhl2, Wnt8b*), and rhombomere 1 (*Cldn10, Fgf8, Fgf17, Spry2*; *Figure 5D*, *Supplementary file 6*). In addition, this distance cutoff assigned midline transcripts in the midbrain/r1 region (*Foxa1, Foxa2*) to a different cluster from midline transcripts in the forebrain (*Gsc, Nkx2-1, Nkx2-4*), and separately clustered lateral transcripts in the midbrain/r1 region (*Pax5, Pax8*) from lateral transcripts that were more broadly expressed along the anterior-posterior axis (*Msx1, Msx2, Pax7, Zic1, Zic5*; *Figure 5D*, *Supplementary file 6*). By contrast, a smaller distance cutoff (higher clustering stringency) of D=4 was

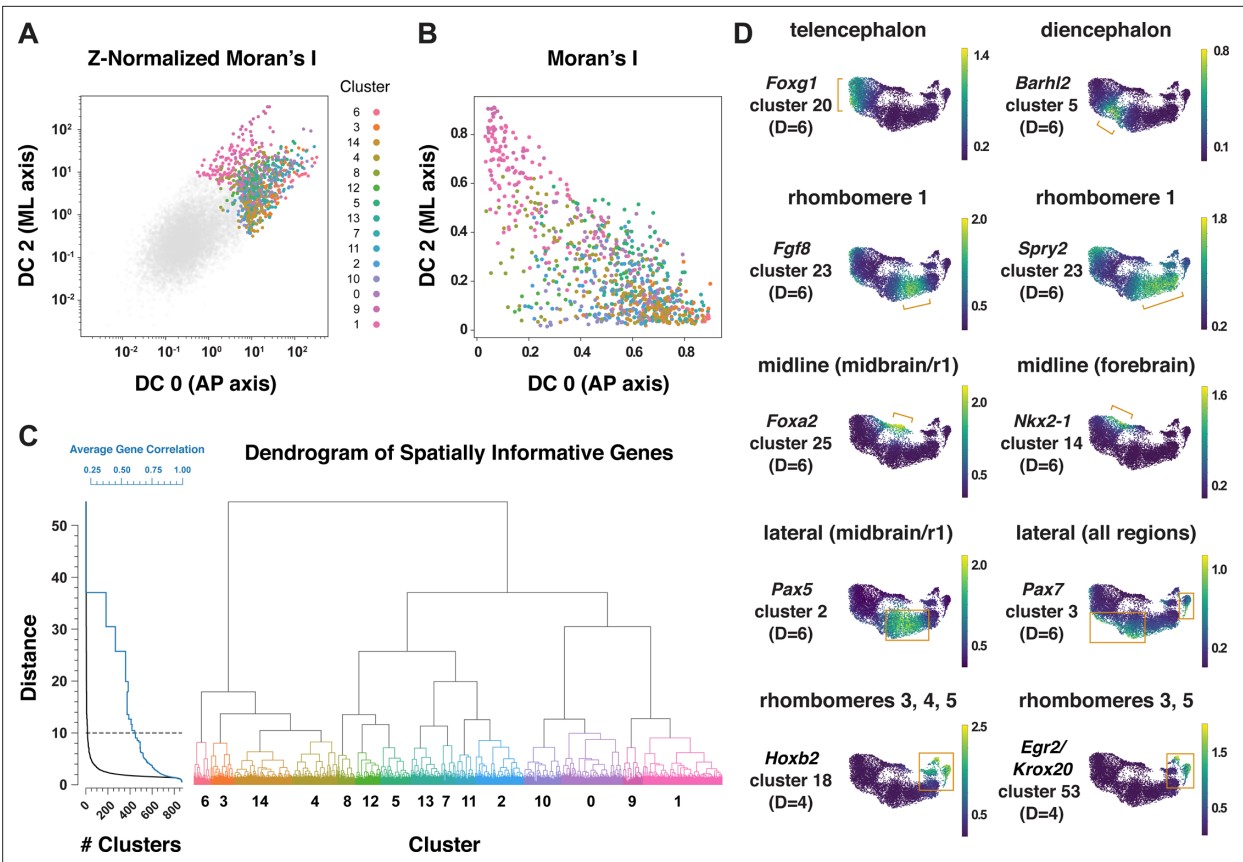

**Figure 5.** Multiscale analysis of spatial gene expression in the mouse cranial neural plate. (**A**) Plot of genes by Z-Normalized Moran's I relative to DC0 and DC2 for all genes expressed in the E8.5–9.0 cranial neural plate. Genes meeting the cutoff for further analysis are colored according to cluster identity at D=10; other genes are shown in gray. (**B**) Spatially informative genes from (**A**) replotted by spatial autocorrelation (Moran's I) along DC0 and DC2, colored by cluster identity at D=10. (**C**) Dendrogram of spatially informative genes showing gene clusters at different distances corresponding to the average correlation among genes. Left, the number of clusters at each distance (black curve) and the average correlation between expression patterns within clusters (blue curve). Right, dendrogram showing the relationships between clusters, colored by cluster identity at D=10 as in A and B. (**D**) UMAP projections of E8.5–9.0 neural plate cells colored by normalized gene expression. Examples of genes that mark specific presumptive brain structures (the telencephalon, diencephalon, and rhombomere 1), midline or lateral domains, or subsets of rhombomeres are shown. Brackets and boxes indicate regions of increased gene expression. Spatial cluster identities at a stringency of D=6 (top and middle panels) or D=4 (bottom panels) are indicated.

The online version of this article includes the following figure supplement(s) for figure 5:

**Figure supplement 1.** Multiscale clustering analysis reveals spatial patterns of gene expression in the cranial neural plate.

**Figure supplement 2.** Comparison of mediolateral patterning in the cranial neural plate and dorsal-ventral patterning in the spinal cord.

necessary to distinguish transcripts specific to rhombomeres 3 and 5 (*Egr2/Krox20*) or rhombomere 4 (*Hoxb1*) from general rhombomere markers (*Hoxa2, Hoxb2*; **Figure 5D**, **Supplementary file 6**). Therefore, clustering genes at varying degrees of stringency provides an effective strategy to distinguish genes with different two-dimensional expression patterns.

We next asked if this multidimensional approach captures aspects of spatial patterning that are detected in other regions of the neural plate in the literature. In particular, markers that define distinct neural precursor populations have been shown to be patterned along the dorsal-ventral axis of the mammalian spinal cord (**Delile et al., 2019**; **Rayon et al., 2021**). To ask if these markers display similar patterns in the cranial region, we analyzed 29 genes expressed in different sets of neural precursors along the dorsal-ventral axis of the spinal cord (**Delile et al., 2019**). Of these, 9 were not expressed in the midbrain/r1 region, and 19 of the remaining 20 genes were predicted to be spatially patterned in our analysis. Of these, 17 genes were expressed in mediolateral domains that were roughly analogous to their positions in the spinal cord (**Figure 5—figure supplement 2**). Notably, the majority of these genes were predicted to be patterned along both mediolateral and anterior-posterior axes,

suggesting that neural precursor transcripts arrayed along a one-dimensional axis in the developing spinal cord are often patterned in two dimensions in the cranial neural plate.

## Spatially regulated expression of genes involved in cell-cell communication

Cranial neural tube morphogenesis and closure require short- and long-range communication between cells mediated by secreted and transmembrane proteins (*Harris and Juriloff, 2007*; *Harris and Juriloff, 2010*). These include short-range interactions mediated by Cdh2/N-cadherin (*Radice et al., 1997*) as well as long-range signaling by the secreted SHH, BMP, and WNT proteins (*Liu and Joyner, 2001a*; *Wurst and Bally-Cuif, 2001*; *Murdoch and Copp, 2010*; *Brooks et al., 2020*; *Kicheva and Briscoe, 2023*). However, the full range of surface-associated and secreted signals that mediate communication between cells in the cranial neural plate has not been systematically characterized. To elucidate mechanisms of spatially regulated cell-cell communication, we applied the mutual information framework to compare genes that display high information content along DC0 or DC2 with ligands and receptors in the Cellinker (*Zhang et al., 2021*) and CellTalkDB (*Shao et al., 2020*) databases and secreted and transmembrane proteins identified by sequence homology (Materials and methods). Using this approach, we identified 194 secreted or transmembrane proteins predicted to be patterned along the anterior-posterior or mediolateral axes or expressed in more complex patterns in the cranial neural plate (104 shown in *Figure 6A–C*, see *Supplementary file 9* for full list).

This analysis identified genes encoding ligands and receptors known to have patterning or morphogenetic functions, such as *Shh* and its inhibitory receptor *Ptch1*, *Wnt* family ligands and their corresponding *Frizzled* receptors, *Fgf* ligands and receptors, and *Ephrin* ligands and their *Eph* receptors (*Figure 6D–F*, *Figure 6—figure supplement 1A–E*). This analysis also predicted patterned expression of genes required for cranial neural tube closure, such as *Celsr1* in the midbrain/r1 and hindbrain and *Vangl1* in the midline (*Figure 6A and B*; *Curtin et al., 2003*; *Torban et al., 2008*). In addition to genes involved in cell-cell communication, we also used this approach to predict the expression of transcriptional regulators in the *Fox*, *Hes*, *Irx*, *Msx*, *Pax*, and *Zic* families (*Figure 6—figure supplement 1F–K*). Intersecting molecularly defined gene families with the spatial map predicted by scRNA sequencing may be a generally useful approach for the targeted investigation of gene families that share sequence features but display a complex spatial relationship.

## SHH signaling promotes distinct transcriptional programs in different cranial regions

Cell identity in the cranial neural plate is spatially and temporally regulated by conserved morphogens and growth factor signaling pathways, but how these signals influence the transcriptional programs that promote cranial neural tube patterning and morphogenesis is not well understood. SHH is an essential regulator of mediolateral and dorsal-ventral patterning in the developing brain and spinal cord. In the spinal cord, loss of SHH signaling reduces or eliminates specific medial cell types, whereas excessive SHH signaling results in an expansion of medial populations at the expense of lateral cell fates (*Ingham and McMahon, 2001*; *McMahon et al., 2003*; *Dessaud et al., 2008*; *Ribes and Briscoe, 2009*; *Goetz and Anderson, 2010*; *Kicheva and Briscoe, 2023*). In cranial tissues, increased SHH signaling interferes with the patterned cell behaviors required for cranial neural tube closure (*Murdoch and Copp, 2010*; *Brooks et al., 2020*), and regulation of SHH is important for determining spatially delimited neural fates (*Fuccillo et al., 2004*; *Blaess et al., 2006*; *Balordi and Fishell, 2007*; *Xu et al., 2010*; *Tang et al., 2013*). However, the transcriptional programs activated by ectopic SHH signaling in the cranial neural plate have not been systematically examined.

To characterize the gene expression changes that occur in response to increased SHH signaling in the cranial neural plate, we performed scRNA-seq analysis of embryos cultured with the small molecule Smoothened agonist SAG, which promotes ligand-independent activation of the SHH receptor Smoothened (*Chen et al., 2002*). Embryos cultured with SAG for 12 hr between E8.0 and E8.5 displayed a lateral expansion of the SHH target genes *Foxa2* and *Nkx6-1* (*Figure 7A*), and disrupted neural fold elevation, consistent with an increase in SHH pathway activity (*Brooks et al., 2020*). Transcriptional profiles were obtained for 4024 and 4140 cells from dissected cranial regions of control and SAG-treated embryos, respectively, with a median library depth of >16,000 transcripts (UMIs)/cell and >4300 genes/cell (average 1.57 reads/UMI) (*Figure 7B*, *Supplementary file 1*). Of these, we

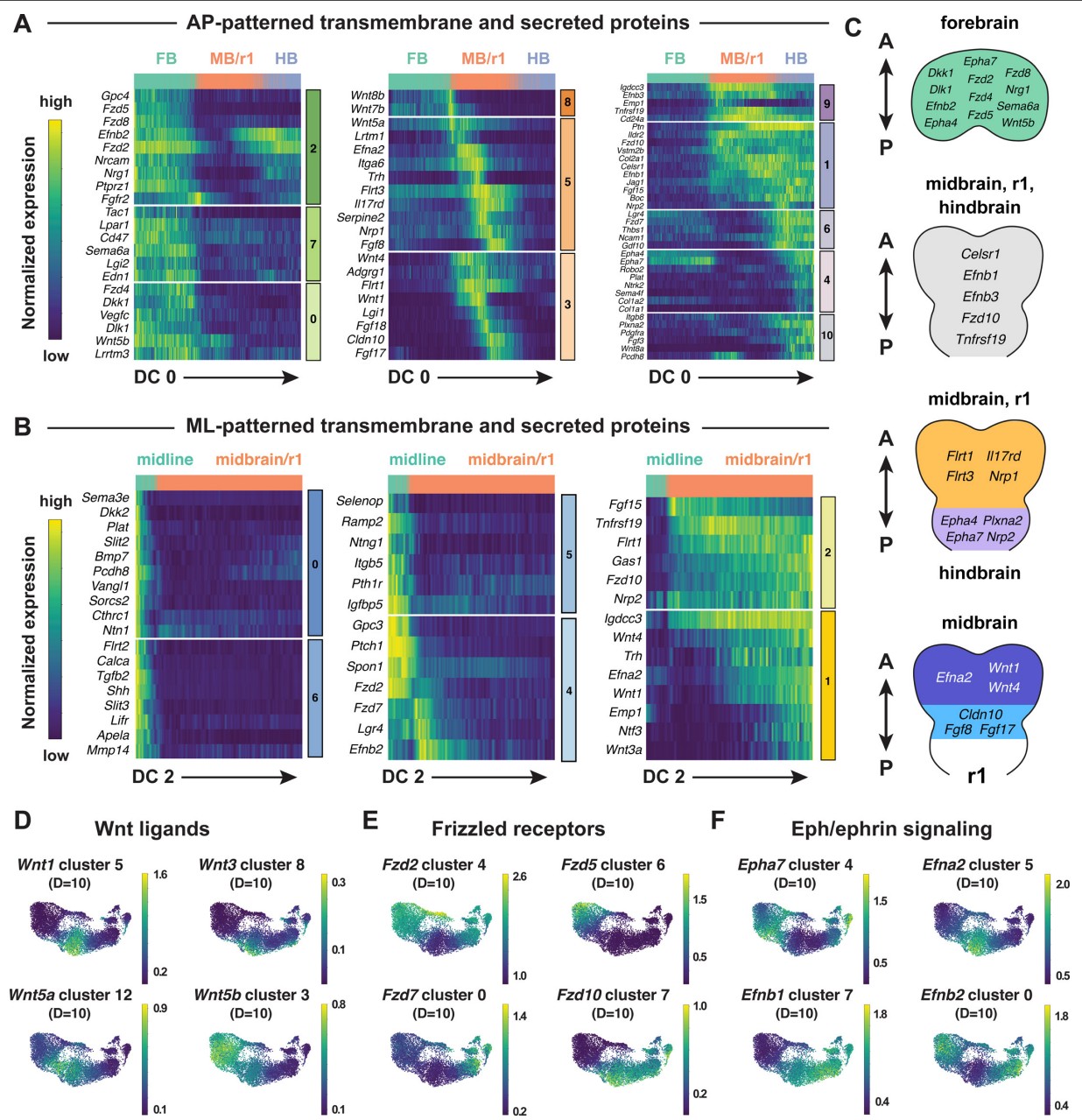

**Figure 6.** Patterned expression of secreted and transmembrane proteins in the cranial neural plate. (**A, B**) Heatmaps showing the normalized expression of transmembrane and secreted proteins relative to DC0 (**A**) or DC2 (**B**). Heatmaps show one gene per row, one cell per column, with the cells in each row ordered by their value along DC0 or DC2. Colored bars (right) show the cluster identity relative to DC0 or DC2. Cells assigned to the forebrain (FB), midbrain/r1 (MB/r1), hindbrain (HB), or midline are indicated at the top of each heatmap. (**C**) Schematics showing the predicted expression of example transmembrane and secreted proteins along the anterior-posterior axis. Anterior (A), posterior (P). (**D–F**) UMAP projections of cranial neural plate cells from E8.5–9.0 embryos colored by the normalized expression of a subset of *Wnt* ligands (**D**), *Frizzled* receptors (**E**), and *Ephrin* ligands and *Eph* receptors (**F**). Spatial cluster identities at D=10 are indicated.

The online version of this article includes the following figure supplement(s) for figure 6:

**Figure supplement 1.** Patterned expression of gene families.

obtained 1619 neural plate cells from control embryos and 1,401 neural plate cells from SAG-treated embryos. These cells were assigned to the forebrain, midbrain/r1, or hindbrain regions using known markers and differentially expressed genes were identified in each region (***Figure 7C–J***, ***Supplementary file 10***). Differentially expressed genes were identified by MAST analysis (***Finak et al., 2015***),

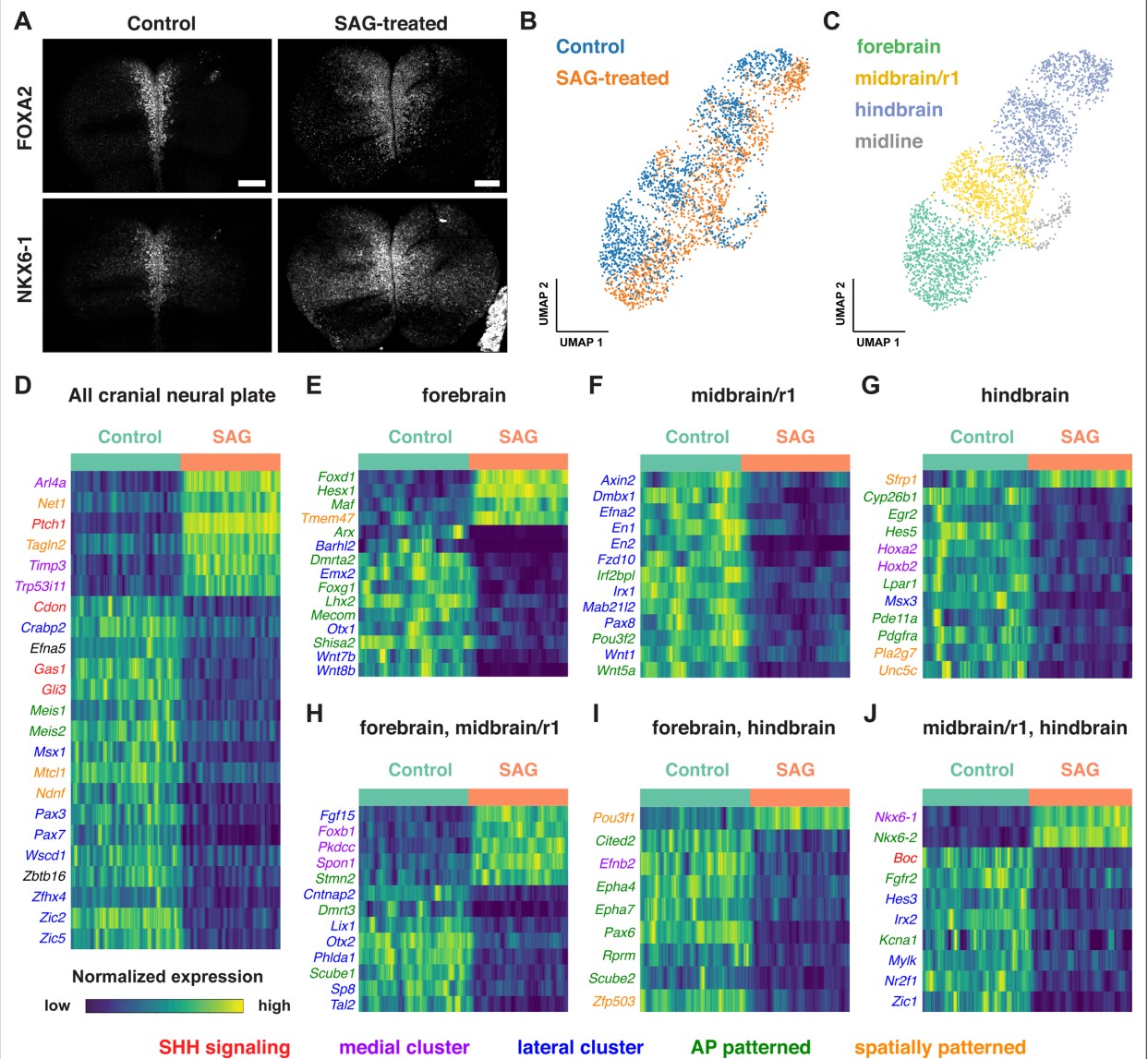

**Figure 7.** Single-cell RNA sequencing reveals region-specific transcriptional programs activated by SHH signaling. (**A**) Maximum-intensity projections of the midbrain and anterior hindbrain regions of the cranial neural plate of 5-somite mouse embryos treated with vehicle control (left) or with 2 μM Smoothened Agonist (SAG) stained for FOXA2 and NKX6-1. SAG-treated embryos display an expanded floor plate region. Bars, 100 μm. (**B, C**) UMAP projections of 1619 and 1409 cranial neural plate cells from control and SAG-treated embryos at E8.5, respectively, colored by treatment (**B**) or neural plate domain (**C**). (**D–J**) Heatmaps showing the normalized expression of candidate SHH target genes that were upregulated or downregulated throughout the cranial neural plate (**D**) or in the indicated region(s) (**E–J**) in SAG-treated embryos. Heatmaps show one gene per row, one cell per column, with the cells in each row grouped by control (green) or SAG treatment (orange), indicated at the top of each heatmap. Genes involved in SHH signaling (red) and genes assigned to medial (purple), lateral (blue), anterior-posterior (green), or other spatial clusters (orange) are indicated. Genes that had a MAST hurdle value >0.24 or<−0.24 and a false discovery rate adjusted p-value of p<0.001 in at least one region were indicated as differentially expressed in all regions in which p<0.001 and the MAST hurdle value was >0.10 or<−0.10.

which accounts for sample variation in scRNA-seq data when comparing log-transformed data (Materials and methods). Using a false discovery rate-adjusted p-value of <0.001, and a MAST hurdle transformation of the $\log_2$ fold-change value of >0.24 or<−0.24, we identified 166 genes that were significantly upregulated and 199 genes that were significantly downregulated in at least one region of the cranial neural plate in SAG-treated embryos (*Supplementary file 11*). Due to the length of SAG treatment, these are predicted to include direct targets of SHH signaling as well as secondary gene expression changes.

Several known SHH pathway genes were deregulated in SAG-treated embryos (*Figure 7D–J*), consistent with an expansion of SHH signaling. The SHH target gene *Ptch1* (*Goodrich et al., 1996*) was one of the top two upregulated genes in all regions of the cranial neural plate in SAG-treated embryos (*Supplementary file 10*, *Supplementary file 11*). In addition, the SHH targets *Gli1* (*Lee et al., 1997*) and *Foxa2* (*Sasaki et al., 1997*) were significantly upregulated but did not meet our MAST hurdle threshold, possibly due to the exclusion of midline cells from our analysis, suggesting that additional SHH targets are also captured in this dataset. In addition to genes that were upregulated in SAG-treated embryos, several genes that are negatively regulated by SHH in other contexts were transcriptionally downregulated in the cranial neural plate, including the SHH co-receptors *Cdon* and *Gas1* and the transcriptional repressor *Gli3* (*Marigo et al., 1996*; *Wang et al., 2000*; *Okada et al., 2006*; *Tenzen et al., 2006*; *Yao et al., 2006*; *Zhang et al., 2006*; *Allen et al., 2007*; *Goetz and Anderson, 2010*). SAG-treated embryos also displayed reduced expression of *Scube2*, which encodes a cell surface protein that modulates SHH secretion (*Woods and Talbot, 2005*; *Creanga et al., 2012*), consistent with recently reported effects of SHH on *Scube2* expression in the zebrafish neural tube (*Collins et al., 2024*). Therefore, our dataset identifies known components of the SHH pathway as well as components of transcriptional feedback mechanisms that modulate SHH signaling.

Notably, over 40% of the 365 genes that were modulated by SAG treatment were also predicted to be spatially regulated in the wild-type cranial neural plate (*Figure 7D–J*), consistent with a strong effect of SHH on spatial patterning. This includes 75 transcripts that were patterned along both anterior-posterior and mediolateral axes, 63 that were patterned along the anterior-posterior axis, and 14 that were patterned along the mediolateral axis (*Supplementary file 11*). As expected, SAG treatment led to the upregulation of genes associated with medial clusters in wild type and the downregulation of genes associated with lateral clusters (*Figure 7D–J*), similar to the effects of SHH in the spinal cord (*Dessaud et al., 2008*; *Ribes and Briscoe, 2009*; *Goetz and Anderson, 2010*). In addition, partially distinct transcripts were upregulated in different anterior-posterior domains in SAG-treated embryos, suggesting a strong convolution between anterior-posterior and mediolateral patterning systems. For example, *Foxd1, Hesx1*, and *Maf* were specifically upregulated in the forebrain (*Figure 7E*), *Foxb1* was upregulated in the forebrain and midbrain/r1 (*Figure 7H*), and *Nkx6-1* and *Nkx6-2* were upregulated in the midbrain/r1 and hindbrain (*Figure 7J*). Moreover, genes that were downregulated in SAG-treated embryos include lateral transcripts present throughout the cranial neural plate (*Msx1, Pax3, Pax7, Zic2, and Zic5*), as well as lateral transcripts expressed in specific regions, such as *Pax8* in the midbrain/r1, *Msx3* in the hindbrain, and *Zic1* in both domains. The region-specific transcriptional changes induced by activated SHH signaling are consistent with the results of our spatial analysis, indicating that many genes are responsive to both anterior-posterior and mediolateral inputs (*Figure 5*, *Supplementary file 6*). Together, these results suggest significant cross-regulation between the mechanisms that define anterior-posterior and mediolateral cell identities.

Activation of SHH signaling by SAG also modified the expression of genes involved in other signaling systems that regulate neural patterning and tissue development. These include factors involved in retinoic acid signaling (*Crabp2* and *Cyp26b1*) and components of the WNT signaling pathway, which has been shown to function antagonistically to SHH in several contexts (*Dessaud et al., 2008*; *Ribes and Briscoe, 2009*; *Ulloa and Martí, 2010*; *Borday et al., 2012*). In particular, several WNT ligands were downregulated in the forebrain (*Wnt7b* and *Wnt8b*) or midbrain/r1 (*Wnt1* and *Wnt5a*) regions of SAG-treated embryos (*Figure 7E and F*), and the WNT effectors *Axin2* and *Fzd10* were downregulated in the midbrain/r1 region (*Figure 7F*). Conversely, the WNT inhibitor *Sfrp1* was upregulated in the hindbrain (*Figure 7G*). These results suggest that SHH could antagonize WNT signaling by both inhibiting the expression of pathway components and enhancing the expression of a pathway repressor. Finally, SAG-treated embryos also displayed a downregulation of transcripts associated with neuronal function, including genes associated with neurodevelopmental disorders (*Irf2bpl*; *Marcogliese et al., 2018*), epilepsy and autism spectrum disorders (*Cntnap2*; *Peñagarikano et al., 2011*), epilepsy and ataxia (*Kcna1*; *Smart et al., 1998*), and GABAergic neuron development and function (*Tal2*; *Bucher et al., 2000*; *Achim et al., 2013*). These results provide clues to the molecular basis of the antagonism between SHH and WNT signaling and suggest potential connections between SHH signaling and human disease.

## Discussion

In this study, we describe a spatially and temporally resolved dataset of single-cell gene expression profiles in the cranial neural plate of the mouse embryo during cranial neural tube patterning and closure. Analysis of gene expression in cranial neuroepithelial cells revealed dynamic changes in gene expression over a 1.5-day window of development that captures the emergence of distinct transcriptional signatures in the future forebrain, midbrain, hindbrain, and midline, allowing the imputation of developmental trajectories in specific spatial domains. In addition, we used this single-cell dataset to reconstruct spatial patterns of gene expression during the final stages of neural tube closure, with computational ordering based on diffusion components providing a good approximation of the anterior-posterior and mediolateral axes. The spatial patterns predicted by our analysis displayed >85% agreement with prior gene expression studies, indicating that this approach can be used to predict the expression of previously unexplored genes. By extending this analysis to multiple dimensions, we identified refined transcriptional domains that integrate inputs from both anterior-posterior and mediolateral patterning systems. This dataset provides a comprehensive spatiotemporal atlas of transcriptional cell states during a critical window of development, revealing the complex spatial and temporal regulation of gene expression that accompanies the morphogenesis and patterning of the cranial neural plate during neural tube closure.

The present study expands on previous scRNA-seq datasets that capture broad spatial identities (*Pijuan-Sala et al., 2019*; *La Manno et al., 2021*; *Zalc et al., 2021*) or elements of anterior-posterior and mediolateral organization (*Lohoff et al., 2022*; *Qiu et al., 2022*; *Qiu et al., 2024*) in the mouse cranial neural plate. Because cell states are strongly dependent on the location of cells relative to signaling centers or other tissue features, the high density of spatial sampling provided by our dataset made it possible to define a diffusion component spanning the entire anterior-posterior axis of the cranial neural plate, as well as a diffusion component that recapitulates the mediolateral organization of cells in the midbrain and rhombomere 1. Other diffusion components identified in this analysis may be useful for exploring different aspects of spatial organization, such as mediolateral patterning in the forebrain (DC6, DC7) and rhombomeres (DC8), rhombomere-specific transcriptional programs (DC3-DC5), and gene expression signatures shared by spatially separated domains (DC6, DC8, DC9). Diffusion-based approaches have been used to define spatial and temporal cell orderings in other tissues, including the mouse embryonic endoderm (*Nowotschin et al., 2019*) and hematopoietic lineage (*Setty et al., 2019*). By combining diffusion component analysis with methods to interrogate spatial autocorrelation (*DeTomaso et al., 2019*; *DeTomaso and Yosef, 2021*), we were also able to identify genes expressed in two-dimensional patterns, an approach that may prove useful in other contexts. Although spatial transcriptomics is important for interrogating gene expression patterns in complex and heterogeneous samples such as tumors (*Longo et al., 2021*; *Rao et al., 2021*; *Moffitt et al., 2022*; *Moses and Pachter, 2022*), our findings suggest that for well-ordered tissues, increasing the sampling density can provide insight into the spatial and temporal dynamics of cell state and gene expression changes without the need for explicitly spatial approaches.

Comparison of gene expression in different regions of the neural plate suggests a general correspondence between dorsal-ventral patterning of neural precursors in the spinal neural tube and mediolateral patterning in the cranial neural plate. However, in contrast to the discrete, sharply delineated transcriptional domains associated with distinct neuronal populations along the dorsal-ventral axis of the spinal cord after closure (*Delile et al., 2019*; *Rayon et al., 2021*), our data suggest less refinement of these domains along the mediolateral axis of the cranial neural plate. At least two factors could contribute to these differences. First, our dataset focuses on early stages of patterning in the cranial neuroepithelium, when cells are likely to share expression of genes that specify a common neuroepithelial identity. Second, these stages may predate the establishment of secondary feedback systems that lead to fine-scale patterning of mutually exclusive neural precursor domains. The continuous nature of gene expression in the cranial neural plate may explain why diffusion-based approaches effectively captured gene expression patterns along the anterior-posterior and mediolateral axes.

We took advantage of the ability to detect spatial gene expression in the cranial neural plate to investigate region-specific transcriptional outcomes of activated SHH signaling, which disrupts neural tube patterning and closure (*Dessaud et al., 2008*; *Goetz and Anderson, 2010*; *Murdoch and Copp, 2010*). These results support a critical role for SHH in mediolateral patterning in the cranial neural plate, as ectopic SHH signaling led to an upregulation of medially expressed genes and a

downregulation of lateral genes, and identify transcriptional changes that could underlie the functional antagonism between the SHH and WNT signaling pathways, which have opposing effects on neural patterning (*Ulloa and Martí, 2010*; *Borday et al., 2012*). Our analysis suggests that crosstalk between the SHH and WNT pathways could involve transcriptional repression of WNT ligands and effectors as well as transcriptional activation of the pathway inhibitory factor *Sfrp1*, perhaps through a GLI-binding site in the *Sfrp1* promoter (*Katoh and Katoh, 2006*). Consistent with this possibility, bulk RNA sequencing of E9.5 mouse embryos lacking the ciliary G-protein-coupled receptor Gpr161, which display expanded SHH signaling (*Mukhopadhyay et al., 2013*), reveals reduced expression of WNT pathway genes, including several that overlap with our dataset (*Kim et al., 2019*). These results suggest that SHH activity could broadly antagonize WNT signaling at the transcriptional level.

Gene expression changed significantly along both mediolateral and anterior-posterior axes in SAG-treated embryos, raising the question of how SHH simultaneously influences both aspects of patterning in the cranial neural plate. In one model, SHH activation could disrupt anterior-posterior patterning in part by inhibiting WNT and retinoic acid signaling, which are required to establish posterior neural fates (*Yamaguchi, 2001*; *Maden, 2007*). Alternatively, expanded SHH signaling may override anterior-posterior gene expression programs in favor of a mutually exclusive medial cell fate. Finally, SHH signaling could intersect with localized regulators to activate distinct gene expression programs in the forebrain, midbrain, and hindbrain, allowing a single morphogen to generate region-specific cell fates and tissue structures. Several mechanisms have been proposed to influence the targets of SHH signaling, including the presence of intermediary or accessory factors such as FOXC1 and FOXC2 in pharyngeal tissues (*Yamagishi et al., 2003*), SOX2 in the spinal cord (*Peterson et al., 2012*), and HAND2 in mandibular development (*Elliott et al., 2020*). In other cases, the SHH signaling response is modulated by the receptivity of cells to SHH, such as RFX4-dependent primary cilia formation in the central nervous system (*Ashique et al., 2009*) or the chromatin accessibility of SHH target sites in the limb bud (*Lex et al., 2020*). The activation of tissue-specific gene expression programs by SHH has also been observed in the chick coelomic epithelium (*Arraf et al., 2020*), raising the possibility that interaction with orthogonal patterning systems may be a general mechanism that shapes the outcome of SHH signaling. Functional analyses of genes predicted to be spatially patterned by scRNA-seq analysis will be necessary to define their developmental contributions and determine their relationship to SHH signaling and other patterning systems in the cranial neural plate.

The systematic profiling of gene expression in the cranial neural plate provides a resource for defining the transcriptional changes that underlie the acquisition of region-specific cell fates in the developing forebrain, midbrain, and hindbrain. These data can provide insight into cell differentiation mechanisms in vivo and inform efforts to induce the differentiation of specific cranial neuronal and neuron-adjacent cell populations from pluripotent stem cells in vitro (*Tabar and Studer, 2014*; *Medina-Cano et al., 2022*). Although the present analysis focused on the cranial neural plate, this dataset will be useful for analyzing dynamic changes in gene expression in other cranial cell populations that exhibit significant changes in transcriptional state. For example, this dataset includes nearly 14,000 cranial mesoderm cells that could shed light on critical roles of the mesoderm during cranial development (*Chen and Behringer, 1995*; *Camus et al., 2000*; *Zohn et al., 2007*; *Zohn and Sarkar, 2012*; *Bildsoe et al., 2013*; *Qiu et al., 2022*; *Qiu et al., 2024*), as well as cells of the cranial neural crest (*Zalc et al., 2021*; *Williams et al., 2022*) and non-neural ectoderm, which contains the cranial placodes (*Koontz et al., 2023*) and contributes essential signals and forces required for cranial neural tube closure (*Dickinson et al., 1995*; *Shimamura and Rubenstein, 1997*; *Molè et al., 2020*; *Maniou et al., 2021*; *Christodoulou and Skourides, 2022*). Further analysis of gene expression during cranial development will provide insight into the spatiotemporal delineation of diverse cell populations during neural tube closure and will provide a framework for systematically elucidating the transcriptional changes that contribute to the morphogenesis and patterning of the mammalian brain.

## Materials and methods
### Mouse handling and use
Mice used in this study were 8- to 12-week-old inbred FVB/NJ mice purchased from the Jackson Laboratory (strain #001800). Timed pregnant dams were euthanized by carbon dioxide inhalation followed by secondary cervical dislocation. All mice were bred and housed in accordance with PHS guidelines

and the NIH Guide for the Care and Use of Laboratory Animals and an approved Institutional Animal Care and Use Committee protocol (15-08-013) of Memorial Sloan Kettering Cancer Center.

## Embryo collection and cell dissociation

FVB embryos were dissected from the uterine horn on ice in pre-chilled DMEM/F-12 with Glutamax (Thermo Fisher), hereafter referred to as DMEM, including resection of the yolk sac and amnion. Embryo stages were assigned as follows: 0 somites (E7.5), 1–2 somites (E7.75), 3 somites (E8.0), 4–6 somites (E8.25), 7–9 somites (E8.5), 10+somites (E8.75-E9.0). Cranial tissues from E7.5-E8.0 embryos were collected by dissecting at the posterior limit of the cranial neural plate. Cranial tissues from E8.25-E9.0 embryos were collected by dissecting embryos just posterior to the otic sulcus. Due to differences in embryo size, a larger region of the hindbrain was included at earlier stages, which may account for a subset of transcripts that were present in early but not late hindbrain samples. Visible cardiac structures, including the early heart tube, were manually resected. Cranial tissues from three to five embryos were pooled per replicate; four to six replicates were analyzed per stage in wild-type embryos and three SAG-treated replicates and two control replicates were analyzed for the SAG treatment experiment (*Supplementary file 1*). Individual cells were dissociated by transferring pooled cranial tissues into 2 mL of pre-chilled phosphate buffered saline (PBS) containing 2.5% pancreatin (Sigma) and 0.5% trypsin (Sigma) and incubated 5 min on ice. Tissues were then washed 1 min in cold DMEM with 10% calf serum (Gibco), followed by a 1 min wash in cold DMEM alone. Pooled tissues were then transferred to a clean watch glass containing 200 µL of a 1:2 mixture of accutase (Sigma) and 0.25% trypsin in PBS and incubated 15 min at 37 °C, gently swirled, and incubated for an additional 15 min. After incubation, tissues were returned to ice and 600 µL of DMEM with 20% calf serum in DMEM was added to each watch glass. Tissues were manually triturated on ice using tungsten needles for 5–10 min, then passed over a 40 µm FlowMi cell strainer (Sigma) and concentrated by $450 \times g$ centrifugation at 4 °C. After centrifugation, the supernatant was removed and the cells were resuspended in 50 µL PBS with 0.4% bovine serum albumin (BSA). 10 µL of cells were combined with 10 µL of Trypan blue and placed in a hemocytometer to analyze the proportion of single cells and dead cells. Pools with less than 85% singlets or 85% viable cells were discarded. Isolated cells were then encapsulated and barcoded using standard protocols on the Chromium V3 chemistry platform from 10X Genomics. 3′ RNA-seq libraries were subsequently generated following standard 10X Genomics protocols.

## SAG treatment

For treatment with Smoothened Agonist (SAG; Sigma 912545-86-9), FVB embryos were dissected at E8.0 for culture following standard protocols in a 50:50 mixture of pre-warmed and gassed (37 °C, 5% $CO_2$) DMEM and whole embryo culture rat serum (Inotiv). After dissection, embryos were randomly assigned to treatment with 2 µM SAG in DMSO or an equivalent volume (0.02% v/v) of DMSO alone as a control and cultured 12 hr in pre-warmed and gassed (37 °C, 5% $CO_2$) DMEM in 24-well Lumox plates (Sarstedt). After the culture period, rare embryos with signs of excessive cell death or developmental failure were discarded; the frequency of these embryos did not differ between control and SAG-treated conditions. After final resection of the yolk sac and amnion, embryo dissection, cell dissociation, and library preparation were performed as described.

## Single-cell transcriptome sequencing

For replicates 3–23 and 40–43 (*Supplementary file 1*), cell suspensions were loaded on a Chromium instrument (10X Genomics) following the user guide manual for 3′ v3. In brief, cells were washed once with PBS containing 1% bovine serum albumin (BSA) and resuspended in PBS containing 1% BSA to a final concentration of 700–1300 cells/µL. The viability of cells was above 80%, as confirmed with 0.2% (w/v) Trypan Blue staining (Countess II). Cells were captured in droplets. Following reverse transcription and cell barcoding in droplets, emulsions were broken and cDNA purified using Dynabeads MyOne SILANE followed by PCR amplification following the manual instructions.

For replicates 44–55 (*Supplementary file 1*), single cell suspensions were stained with Trypan blue and Countess II Automated Cell Counter (ThermoFisher) was used to assess both cell number and viability. Following quality control, the single cell suspension was loaded onto Chromium Next GEM Chip G (10X Genomics PN 1000120) and GEM generation, cDNA synthesis, cDNA amplification,

and library preparation of up to 5000 cells proceeded using the Chromium Next GEM Single Cell 3' Kit v3.1 (10X Genomics PN 1000268) according to the manufacturer's protocol. cDNA amplification included 11 cycles and 17–265 ng of the material was used to prepare sequencing libraries with 8–14 cycles of PCR.

After PicoGreen quantification and quality control by Agilent TapeStation, indexed libraries were pooled equimolar and sequenced on a NovaSeq 6000 in a PE28/91 or PE28/90 run, using the NovaSeq 6000 SP, S1, S2, or S4 Reagent Kit (100 or 200 cycles; Illumina).

## FASTQ alignment

FASTQ files were preprocessed using the using the Sequence Quality Control (SEQC) bioinformatics pipeline (*Azizi et al., 2018*) aligning reads to the mm38 mouse reference genome with default parameters for the 10 x single-cell 3-prime library. The SEQC pipeline performs read alignment, multi-mapping read resolution, as well as cell barcode and UMI correction to generate a cells x genes count matrix. The pipeline was run without performing default cell filtering steps (our strategy for cell filtering is described below). Some gene names have been updated in subsequent genome revisions, but for data interoperability we have preserved the original gene symbols.

## Ambient RNA correction and cell filtering

Aligned, unfiltered count matrices of each sample were processed using CellBender (*Fleming et al., 2023*) to remove ambient RNA contamination and identify empty droplets. The expected number of real cells input to CellBender's *remove-background* function was determined by SEQC (which uses the optima in the second derivative of library size to identify real cells from droplets with ambient RNA expression) and the total droplets used to estimate ambient background RNA was set to 30,000. Training was run for 100 epochs. All droplets that were estimated by CellBender with nonzero probability to be real cells were kept for downstream analysis. From the raw count matrix corrected for ambient RNA contamination, cells that did not pass the following filters were removed: (1) Library size >1000 UMIs. (2) Library complexity: we use SEQC's method to remove cells with low ratio of UMIs vs. genes expressed, with a 0.1 cutoff on the residual for the linear model fit. This step primarily removes high library-size cells with one predominant gene program, for example mitochondrial gene-enriched stressed cells. (3) Percent mitochondrial RNA <20%.

After initial filtering, each sample showed a clear bimodality separating cells of high library size and number of genes expressed and cells of low to moderate library size. In addition, none of the cells from replicate 48 passed the filtering criteria and replicate 54 was removed from the dataset due to low library size (<100 UMIs per cell), so these samples are not reported.

We sought to assess whether the resulting bimodal distribution of library size was biologically driven. For this, we first fit a kernel density estimation model to the library size distributions to systematically assign cells to the higher quality (library size and number of genes expressed) or lower quality mode. We then proceeded to compute differentially expressed genes (DEGs) between the two classes. In order to control for the differences in library size between the two groups, we downsampled both groups to 1000 UMIs (the lowest number of UMIs in the low-quality group) and then identified DEGs calculated between the high and low-quality cells using their raw expression matrices (Wilcoxon rank-sum, Benjamini-Hochberg correction). We found that most of the genes differentially upregulated in the low library size group were associated with ribosomal activity or cell stress and cell death (e.g. GM42418, which was the most significantly upregulated). This evidence suggested that this low-quality cluster of cells likely comprised cells that were under stress and perhaps leaking RNA, resulting in a bias to the capture and amplification of highly expressed genes.

In order to systematically remove the above identified low quality cells with lower library size, we finely clustered the cells using PhenoGraph with a value of *k=8* (*Levine et al., 2015*). Cells of each cluster were reassigned as low- or high-quality based on whether the library size and no. genes distributions of their cluster were more likely to come from the high-quality group described above. Because the library sizes were approximately normal for each group and cluster, a Z-test was performed and p-value cutoff of $1e^{-10}$ was used to decide from which group each cluster was more likely to come. The motivation for filtering by cluster, rather than threshold values, was to avoid relying on a single hard cutoff; instead, we retained cells based on phenotypic similarity, which is captured by cell clustering.

In total, 39,463 cells in the wild-type dataset passed quality control filtering, with a median library depth of >42,000 transcripts (UMIs)/cell and >5900 genes/cell (average 1.56 reads/UMI). In addition, 8164 cells in the control and SAG treatment dataset passed quality control filtering, with a median library depth of >16,000 UMIs/cell and >4300 genes/cell (average 1.57 reads/UMI).

## Normalization and preprocessing

All cells within each dataset were first normalized to median library size and the natural log of normalized expression with pseudocount 1 was computed for each cell. Cell cycle influence was removed by regression with the Python package, fscLVM, using several cell-cycle associated GO gene sets provided in *Supplementary file 12*. For each dataset, feature selection was performed using the 'highly_variable_genes' function in scanpy with 'flavor = seurat_v3' and 'n_top_genes = 3000'. All mouse mitochondrial genes, ribosomal genes, and genes expressed in fewer than 10 cells were excluded from feature selection. In addition, genes corresponding to each cell type and spatial axis delineation were compiled from literature and retained. Principal component analysis was performed on the log-transformed normalized expression matrix; the number of principal components was selected to explain 75% of the variance in the dataset. Cells were clustered with PhenoGraph with the number of nearest neighbors set to 30, after confirming the Rand Index is robust for nearby values (RI >0.8 for an absolute difference of 5 and 10 in $k$ values).

## Cell typing

To identify major cell types in these scRNA-seq datasets, first, we manually annotated each PhenoGraph cluster based on the expression of literature derived gene signatures and identification of cell type-specific DEGs. More specifically, the clusters were first annotated through the following steps: (1) We identified cell type signatures enriched in each cluster by computing the average expression of each signature (see 'Computing Gene Set Scores' below) per cluster. The list of signatures compiled from the literature is provided in *Supplementary file 13*. The expression of cell type signatures in the wild-type dataset is shown in *Figure 1—figure supplement 1A*. (2) We examined DEGs per cluster using the R package MAST on $\log_2$-transformed normalized counts. The DEGs of each cluster in the full cranial dataset are listed in *Supplementary file 2*. This provided a preliminary cell type assignment for each cluster.

Because these datasets capture developmental states at which cell types are not fully resolved, unique cell types were not consistently separated by cluster, especially in samples from early time points. For this reason, we refined the cluster-based annotations by implementing a semi-supervised classification using the *classify* method in the PhenoGraph package. Briefly, PhenoGraph classification began with a training set of labeled cells (e.g. whose cell type is known) and a test set of unlabeled cells (e.g. whose cell type is unknown). An absorbing Markov chain was computed from the dataset, and the absorption probabilities for each cell type label were calculated for all unlabeled cells; for each cell type and unlabeled cell, the absorption probability represents the likelihood that a random walk originating from any cell of that cell type will reach the unlabeled cell. All cells were then labeled according to their maximum absorption probability.

Beginning from cluster-based cell type annotations, a labeled training set of cells was created for each cell type using the following method: First, threshold values equal to the 20th percentile cell type signature scores were calculated for all cells previously given a cell type label. Cells whose score exceeded the threshold for their own cell type and were below the thresholds for all other cell types were used as training examples. All other cells were unlabeled and re-annotated using PhenoGraph's *classify* method, with a value of *k=30*. Cell type compositions of each sample are provided in *Supplementary file 1*. After classification in this manner, we discovered one PhenoGraph cluster of cells (cluster 28) which did not express any neural plate cell type markers listed above. However, we found this cluster differentially expressed multiple hemoglobin genes including *Hba-a1*, *Hba-x*, *Hbb-bh1*, *Hbb-y* (*Supplementary file 2*), so we re-annotated this cluster as 'Blood Cells'. Additionally, manual analysis of cluster 12 revealed relatively high levels of *Foxa2* expression and this cluster was retyped as endoderm.

Finally, to identify major spatial domains of the neural plate, each dataset was first subset to all neural plate cells and processed as described above. The same cell typing procedure was performed on the neural-plate-only datasets using cell type signatures compiled from the literature for each

spatial domain (forebrain, midbrain/r1, hindbrain, and midline) and the same input parameters. Cell types in the cranial neural plate were identified using the markers described in *Supplementary file 13* and the DEGs associated with each cluster are listed in *Supplementary file 3*.

## Computing per-cell geneset scores

For each cell type, we computed a signed signature score, as described in *DeTomaso et al., 2019*, which uses sets of genes known to be expressed in that cell type (positive markers) as well as genes which are known not to be expressed in that cell type (negative markers). The score was calculated as the mean difference in expression of positive and negative gene sets, z-normalized using the expected mean and variance of a random signature with the same number of positive/negative genes. Where no negative genes are provided, the geneset score was simply the z-normalized mean expression of genes compared to a random signature with the same number of genes. This method was used to score spatially patterned gene clusters.

## UMAP visualization

Each dataset was embedded in two dimensions with UMAP (*McInnes et al., 2018*), which was run on a k-nearest neighbor graph (k=30) produced from its principal components using the *umap* function in the Scanpy package. The UMAP initializations were based on partition-based graph abstraction implemented in the Scanpy package using PhenoGraph clusters. Calculation of principal components and PhenoGraph clusters is as described above. Gene expression was visualized using imputed values unless otherwise specified. Gene expression was imputed with MAGIC, using parameters k=5 and t=3 (*van Dijk et al., 2018*).

## Temporal analysis

To identify temporal gene expression trends in the neural plate, we first separated our complete neural plate dataset—consisting of all embryo stages–into three subsets, each containing only cells from the forebrain, midbrain/r1, and hindbrain, respectively. This was done in order to minimize the influence of anterior-posterior patterned changes in gene expression during analysis. For each subset, a diffusion map was computed using the implementation in Palantir (*Setty et al., 2019*) on the log-normalized count matrix (natural log with pseudo-count 1). A k=30 value was used to determine the nearest-neighbor graphs. We verified that the first diffusion component (DC0) in each region was associated with changes in somite stage (*Figure 2A–C*). We further observed changes in E-cadherin (*Cdh1*) and N-cadherin (*Cdh2*) expression that were consistent with this interpretation (*Figure 2D–F*).

For each spatial subset of the data, expression trends of temporally regulated genes were computed for DC0 and clustered using Palantir. Briefly, for each gene, a gene trend was calculated by de-noising gene expression using MAGIC imputation and fitting a generalized additive model to its values of expression for cells ordered along a provided axis of change (this step is identical to Palantir except that DC0 was the analogue of the 'pseudotime' axis used in Palantir). Gene trends were clustered with a value of *k=20* to produce 5, 5, and 6 gene trend clusters in the forebrain, midbrain/r1, and hindbrain subsets, respectively.

## Diffusion component analysis

We first sought to establish that cell-cell similarity in spatial position is well-described by the top diffusion components of the developed neural plate dataset. A diffusion map was computed using the implementation in Palantir (*Setty et al., 2019*) on the natural log-normalized count matrix. A k=30 value was used to determine the nearest-neighbor graph. The number of diffusion components was chosen based on the eigengap, selecting the components preceding the largest eigengap found within the first 40 eigenvalues. For the E8.5–9.0 cranial neural plate dataset, this resulted in the selection of 10 diffusion components (*Figure 3—figure supplement 1*).

Each diffusion component was correlated (Pearson correlation) with all genes and the top 100 most correlated genes (*P*-value <0.01) were identified (*Supplementary file 5*). The diffusion components were found to strongly correlate with genes known to be spatially patterned within the neural plate. Using highly correlated genes known from the literature, each diffusion component was annotated with the specific spatial patterns in the neural plate it represents (see below). As diffusion components 0 and 2 were strongly correlated with markers of the anterior-posterior and mediolateral axes,

respectively (*Figures 3 and 4*), we concluded that components 0 and 2 of the diffusion map, selected by eigengap, provide information associated with spatial patterning of the neural plate; for downstream analysis, we used diffusion components 0 and 2 as proxies for the anterior-posterior and mediolateral axes, respectively.

In addition to the analyses of DC0, DC1, and DC2 discussed in the main text, analysis of the top 25 correlated genes revealed distinct spatial patterns for DC3-DC9. Genes expressed in rhombomeres 5 and 6 were positively correlated with DC3 (*Hoxb3, Mafb, Hoxd3, Hoxa3, Egr2*); genes expressed in rhombomeres 3 and 5 were positively correlated with DC4 (*Egr2, Cyp26b1, Hoxb3, Hoxa2, Mafb, Hoxa3*); genes expressed in rhombomeres 1, 3, and 4 were positively correlated with DC5 (*Hoxb1, Foxd3, Epha2, Egr2, Wnt8a, Hoxb2, Fgf8, Fgf17, Otx1, Crabbp2, Mafb*); genes expressed in the forebrain and midbrain/r1 were positively correlated with DC6 (*Pax6, Emx2, Pax2, Pax8, Barhl2, Pax5, Fgf8, Nkx2-1, Six6, Hesx1, En2, En1*); genes expressed in the forebrain and midline were positively correlated with DC7 (*Nkx2-1, Gsc, Fgf8, Foxd1, Nkx2-9, Fexf2, Foxg1, Sp8, Ptch1*); genes expressed in the forebrain and rhombomeres 2–5 were positively correlated with DC8 (*Six6, Foxd1, Dkk1, Vax1, Meis2, Emx2, Lhx5, Epha7, Dmbx1, Foxg1, Barhl2*); and genes expressed in lateral domains of all anterior-posterior regions were positively correlated with DC9 (*Tfap2c, Pax3, Msx2, Msx1, Tfap2a, Zic5, Pax7, Foxb1, Msx3, Zic2*).

## Selection of spatially informative genes using Hotspot

Having established that our diffusion map contains information associated with the spatial positioning of cells in the neural plate, we next sought to perform feature selection to isolate genes that exhibit patterns of expression in physical space (treating the entire diffusion space in this analysis as an approximation of spatial position). Spatially informative genes were selected using the Hotspot procedure, as described in *DeTomaso and Yosef, 2021*. Briefly, Hotspot identifies spatially informative genes in two steps: (1) A similarity graph is computed between cells as a k-nearest neighbor graph in a user-defined space. (2) Feature selection is performed to identify spatially autocorrelated genes, which are genes whose patterns of expression are well-represented by the similarity graph. To do so, we defined a 'local autocorrelation' test statistic evaluated on each gene, defined as the sums of weighted pairwise products of nearby cells in the similarity graph. The resulting values were transformed into $Z$ scores, given a null model in which the expression of a gene by each cell is independent of similar 'local' cells, for significance calculation which we then used to filter genes.

To construct a similarity graph between cells, an embedding was calculated on diffusion components on three latent spaces: (1) A latent space comprising only the anterior-posterior axis-correlated diffusion component (DC0). (2) A latent space comprising only the mediolateral axis-correlated diffusion component (DC2). (3) A multiscale space embedding was calculated on the top 10 diffusion components of the developed neural plate dataset.

These latent spaces relate to the sections 'Modular organization of gene expression along the anterior-posterior axis of the cranial neural plate', 'Patterned gene expression along the mediolateral axis of the midbrain and rhombomere 1', and 'An integrated framework for analyzing cell identity in multiscale space', respectively, within the main text. The three analyses relating to these sections are also elaborated in the two sections which follow. In each case, the Hotspot open-source Python package was used to create a k-nearest neighbor graph on the latent space with a value of *k=30*, and to compute local autocorrelations of all 19,623 genes detected in this scRNA-seq dataset with respect to the kNN graph. Spatially informative genes were selected according to the following criteria: (1) To eliminate lowly expressed genes, all genes were removed whose natural log-transformed range of expression was less than or equal to 1. (2) All genes were kept whose False Discovery Rate (FDR) values, calculated by Hotspot from p-values associated with genes' spatial autocorrelations, were below $10^{-5}$, and whose Z-scored spatial autocorrelation statistic values were greater than or equal to 10. The latter value was chosen by knee point, calculated on all genes.

## Gene trend clustering

We first sought to classify genes within this set based on their primary spatial axis of patterning (anterior-posterior *or* mediolateral), and cluster genes with similar patterns of expression along each axis individually. To identify genes with anterior-posterior axis-dependent patterns of expression, the Hotspot package was run with all genes, a value of *k=30*, and a latent space comprising only the

anterior-posterior axis correlated diffusion component (diffusion component 1); 483 genes, having a Z-scored spatial autocorrelation values calculated on DC0 greater than 10 and FDR values below $10^{-5}$, were considered patterned along the anterior-posterior axis.

Expression trends of anterior-posterior axis-patterned genes were computed for DC0 and clustered using Palantir (*Setty et al., 2019*) Briefly, for each gene, a gene trend was calculated by de-noising gene expression using MAGIC imputation and fitting a generalized additive model to its values of expression for cells ordered along a provided axis of change (this step is identical to Palantir except that DC0 was the analog of the 'pseudotime' axis used in Palantir). Because known anterior-posterior axis patterns of gene expression are not strongly recapitulated in the midline cells of the neural plate, only non-midline cells were used to compute anterior-posterior axis gene trends. Gene trends were clustered with a value of *k=20* to produce 11 gene trend clusters. The value of k was selected as being sufficiently low to distinguish known patterns of gene expression along the anterior-posterior axis.

Mediolateral axis-dependent patterns of gene expression were identified similarly, using the mediolateral axis-correlated diffusion component (DC2), resulting in 253 mediolateral axis-patterned genes with Z-scored autocorrelation values greater than 10 and FDR values below $10^{-5}$. Mediolateral axis gene trends were calculated on midline and midbrain/r1 cells, which most strongly recapitulate mediolateral patterns of gene expression, and 7 gene trend clusters were identified for this axis using a value *k=20*. Visualization of the UMAP representations indicated that cluster 3 in this analysis contained genes that were highly correlated with non-midbrain/r1 anterior-posterior identities; we therefore excluded this cluster from further analyses.

## Identification of gene clusters in multiscale space

Comparing the spatial autocorrelation values of genes in the anterior-posterior and mediolateral axes, a large fraction of genes that were patterned relative to either DC0 or DC2 were also patterned along the orthogonal axis (*Supplementary file 6*), suggesting that these groups contain genes that respond to inputs from both anterior-posterior and mediolateral patterning systems. For this reason, we sought to integrate multiple spatially relevant DCs into one analysis. To do so, a multiscale space embedding was calculated on the top 10 diffusion components of the developed neural plate dataset. We chose to use multiscale space as it provides an appropriate Euclidean space to more accurately measure the phenotypic similarity/distance between cells. Using this latent space, the Hotspot package was run with all genes and a value of *k=30*. Of the 19,623 genes detected in the cranial neural plate scRNA-seq dataset, 870 genes were identified as spatially informative.

Spatially informative genes were grouped into modules by performing hierarchical clustering on the Pearson correlation coefficient matrix. Correlations were calculated from the log-normalized count matrix, and hierarchical clustering was performed using Euclidean distance and the Ward linkage criterion. The number of clusters was determined by a linkage distance threshold, above which clusters were not merged. This value was determined by finding the knee-point on the median inter-cluster pairwise correlation of genes, calculated for increasing numbers of clusters (*Figure 5C*), and corresponds to an average correlation of approximately 0.5 among genes belonging to the same cluster. We chose this method to compromise between the similarity of genes within a cluster and the total number of clusters considered. In total, 15 spatially patterned gene clusters were identified (*Figure 5C*).

## Visualization of gene expression trends

To visualize gene trends along each DC axis, we use generalized additive models (GAMs) with cubic splines as smoothing functions as in Palantir (*Setty et al., 2019*). In our work, trends for a module score were fitted using a regression model on the DC values (x-axis) and module score values (y-axis). The resulting smoothed trend was derived by dividing the data into 500 equally sized bins along each DC and predicting the module score at each bin using the regression fit.

## Validation of predicted gene expression patterns

Prediction of gene expression patterns were performed on cells from E8.5–9.0 embryos based on the relative expression of genes along DC0 (to predict the anterior-posterior expression domain), or DC2 (to predict the mediolateral domain). To compare predictions to known gene expression patterns, we mined the Mouse Genome Informatics Gene Expression Database (http://informatics.jax.org)

for images of RNA expression by in situ hybridization or protein localization by immunostaining of wild type embryos Theiler Stages 12–14. Links to the corresponding images were computationally generated in Python. This code, which can be used to look up available gene expression images for any gene at any stage in the Mouse Genome Informatics Gene Expression Database, is available as an annotated iPython notebook containing the code and instructions on Github (https://github.com/ZallenLab/Brooks-et-al-scRNAseq-analysis-notebook, copy archived at *Brooks, 2025a*; https://github.com/ZallenLab/MGI-Gene-eXpression-Database-search, copy archived at *Brooks, 2025b*). Links to the requested stages are provided even if no images are currently available. For each gene predicted by our analysis to be in an anterior-posterior or mediolateral cluster, we compared the gene expression pattern in the cranial region in the published images to the predicted gene expression and a Yes, No, or NA (not applicable) determination was made. Genes were assigned as Yes if the predicted gene expression domain(s) and no others in cranial tissues showed expression in MGI database. Genes were assigned as No if the gene was not expressed in the predicted region, or if significant expression outside of the predicted domain(s) within the cranial region was present, even if the gene was also expressed in the predicted domain(s). If no suitable pictures were present, the gene was assigned as NA.

Examples of images that match the anterior-posterior patterns predicted by our analysis can be found in the indicated panels in the following references: *de la Pompa et al., 1997* (*Ascl1*, Figure 5C); *Zhao and Duester, 2009* (*Axin2*, Figure 3C); *Tamplin et al., 2011* (*Clu1*, supplementary data); *Vacalla and Theil, 2002* (*Casz1*, Figure 2B); *Hadjantonakis et al., 1998* (*Celsr1*, Figure 2F); *Lyn and Giguère, 1994* (*Crabp1*, Figure 2A); *Tahayato et al., 2003* (*Cyp26c1*, Figure 2A); *Lewis et al., 2007* (*Dkk1*, Figure 1J): *Broccoli et al., 2002* (*Dmbx1*, Figure 2A; *En1*, Figure 2B); *Dunty et al., 2008* (*Dusp6*, Figure 2J); *Flenniken et al., 1996* (*Efnb1*, Figure 5A); *Ishikawa et al., 2003* (*Egr2*, Figure 5I); *Anselme et al., 2007* (*Emx2*, Figure 3O; *Pax6*, Figure 2H); *Hirata et al., 2004* (*Fezf2*, Figure 1A); *Wright and Mansour, 2003* (*Fgf3*, Figure 1A); *Cajal et al., 2012* (*Fgf8*, Figure 9M and *Hesx1*, Figure 9D); *Chung et al., 2006* (*Foxg1*, Figure 4A; *Gbx2*, Figure 2I); *Waters et al., 2003* (*Gbx1*, Figure 2G); *Lobe, 1997* (*Hes3*, Figure 1E); *Bertrand et al., 2011* (*Hoxb1*, Figure 1—figure supplement 1I); *Wong et al., 1999* (*Mab21l2*, Figure 1B); *Oulad-Abdelghani et al., 1997* (*Meis2*, Figure 4B); *Liguori et al., 2003* (*Nkx2-1*, Figure 4B); *Tian et al., 2002* (*Otx1*, Figure 5G); *Furushima et al., 2007* (*Shisa1*, Figure 3B); *Treichel et al., 2001* (*Sp5*, Figure 2F); *Hallonet et al., 1998* (*Vax1*, Figure 3A); *Mielcarek et al., 2009* (*Vgll3*, Figure 3A); *Tamplin et al., 2008* (*Wfdc2*, supplement).

Examples of images that match the mediolateral patterns predicted by our analysis can be found in the indicated panels in the following references: *Lin et al., 2000* (*Arl4*a, Figure 2A); *Tamplin et al., 2011* (*Calca, Spink1, Wif1*, supplementary data); *Durmus et al., 2006* (*Cthrc1*, Figure 1B); *Bouchard et al., 2005* (*En2*, Figure 2C); *Burke and Oliver, 2002* (*Foxa2*, Figure 2C); *Lee and Fan, 2001* (*Gas1*, Figure 1C); *Hui et al., 1994* (*Gli3*, Figure 4B); *Klymiuk et al., 2012* (*Ifitm1*, Figure 2A); *Failli et al., 2002* (*Lmx1a*, Figure 1A); *Foerst-Potts and Sadler, 1997* (*Msx1*, Figure 7A); *Trainor et al., 2002* (*Msx2*, Figure 3D); *Pabst et al., 1998* (*Nkx2-2*, Figure 4C); *Danesh et al., 2009* (*Nog*, Figure 1D); *Shylo et al., 2020* (*Shh*, Figure 6D); *Pryor et al., 2014* (*Vangl1*, Figure 2A).

## Differential gene expression in SAG-treated embryos

For cells belonging to spatial domains of the neural plate (forebrain, midbrain/r1, and hindbrain), DEGs were calculated between SAG-treated and control populations using the R package MAST (*Finak et al., 2015*) on $\log_2$-transformed normalized counts. Midline cells were not included in this analysis, due to the low cell number recovered from both control and SAG treated samples. The DEGs related to each spatial domain for SAG-treated and control embryos are listed in *Supplementary file 10* and *Supplementary file 11*.

## Expression analysis of transcription factors and receptors

For the analysis of transcriptional regulators, spatially patterned genes in the cranial neural plate (*Supplementary file 6*) were compared with a list of *Mus musculus* genes associated with the transcription factor term in the KEGG BRITE database (*Kanehisa et al., 2023*) using Release 110.0 or the transcription regulator activity GO term (GO: 0140110; *Ashburner et al., 2000*; *Aleksander et al., 2023*) using the May, 2022 release (*Supplementary file 8*). *Nrg1, Wnt3a*, and *Wnt4* were excluded based on sequence homology. For the analysis of ligands and receptors, spatially patterned genes in

the cranial neural plate were compared with a list of predicted mouse ligands and receptors from the Cellinker database (*Zhang et al., 2021*) and the CellTalk database (*Shao et al., 2021*; *Supplementary file 9*). *Ada, Bex3, Lman1, Mip,* and *Mylk* were excluded based on sequence homology. Additional predicted ligands (*Lgi1, Lgi2*) and receptors (*Cldn10, Emp1, Lrtm1, Pcdh8, Tnfrsf19,* and *Vangl1*) that did not appear in either database were added.

## Immunofluorescence

Timed pregnant dams were euthanized and embryos collected in ice cold PBS and then fixed in 4% paraformaldehyde (PFA) (Fisher 50-980-494) for 1–2 hr at room temperature or overnight at 4 °C. Embryos were then washed three times for 30 min each in PBS +0.1% Triton X-100 (Fisher 327371000) (PBTriton) at room temperature followed by a 1 hr incubation in blocking solution (PBTriton +3% BSA) at room temperature. Embryos were then incubated in staining solution (PBTriton +1.5% BSA) with primary antibodies overnight at 4 °C, followed by three 30 min washes in PBTriton at room temperature. Embryos were then incubated in staining solution with Alexa Fluor-conjugated secondary antibodies (1:500; Thermo Fisher) for 1 hr at room temperature, followed by three 30 min washes in PBTriton at room temperature. Embryos were stored in PBTriton at 4 °C until imaging. Primary antibodies were rat anti-E-cadherin (1:300; Sigma U3254), rabbit anti-FoxA2 (1:1000; abcam ab108422), rabbit anti N-cadherin (1:500; Cell Signaling Technology D4R1H), and mouse anti-Nkx6-1 (1:50; Developmental Studies Hybridoma Bank F55A10; *Pedersen et al., 2006*). Secondary Alexa fluor conjugated antibodies were added along with Alexa 546-conjugated phalloidin (Molecular Probes) at a 1:1000 dilution. Embryos were washed three times for 30 min each in PBTriton at room temperature and imaged on a Zeiss LSM 900 confocal microscope.

## Confocal imaging

Whole-mount imaging was performed by mounting embryos dorsal side down in PBTriton (PBS +0.1% Triton X-100) in Attofluor cell chambers (Thermo Fisher A7816), using a small fragment of broken coverglass (Corning 2850–18) with small dabs of vacuum grease (Dow Corning DC 976) to mount the embryo on a #1.5 coverglass (Fisher 12-546-2P). Embryos were imaged by confocal microscopy on inverted microscopes on a Zeiss LSM700 equipped with a Plan-NeoFluar 40 x/1.3 objective, a Zeiss LSM900 with a Plan-NeoFluar 40 x/1.3 objective, or a Leica SP8 with an HC PL Apo 40 x/1.3 objective. Images were captured by tile-based acquisition of contiguous z-stacks of 50–150 µm depth with 0.6–1.2 µm optical slices and 0.3–0.5 µm z-steps. For tiled images, computational stitching was performed using 10% overlap per tile in Zen (Zeiss) or LAS-X (Leica) software.

## Figure assembly

Heatmaps, UMAPs, and plots of normalized expression levels were generated from cleaned and normalized expression data using Scanpy software (*Wolf et al., 2018*). For the heatmaps, gene expression was normalized to the highest expression of that gene along the diffusion component or to the highest expression of that gene within the treatment group (control or SAG). Only forebrain, midbrain/r1, and hindbrain cells were included in the E8.5–9.0 DC0 heatmaps (midline cells were excluded). Only midline and midbrain/r1 cells were included in the E8.5–9.0 DC2 heatmaps (forebrain and hindbrain cells were excluded). For the SAG analysis, heatmaps that show cells from specific regions or combinations of regions are indicated.

Maximum-intensity projections of image z-stacks were generated in Zen (Zeiss), LAS-X (Leica), or FIJI. Figures were assembled using Photoshop and Illustrator software (Adobe). Fluorescence images were contrast adjusted for display using the FIJI redistribution of ImageJ (*Schindelin et al., 2012*).

## Acknowledgements

The authors thank Meg Distinti, Tarek Islam, and Matthew Schilling for technical assistance and Alex Joyner, Tom Schultheiss, and members of the Zallen lab for helpful feedback on the manuscript. ERB was supported by NIH/NINDS F32 fellowship NS098832 and startup funds from North Carolina State University. AM, ML, and RS were supported by the Alan and Sandra Gerry Metastasis and Tumor Ecosystems Center, Memorial Sloan Kettering Cancer Center. All sequencing was performed by the Sloan Kettering Integrated Genomics Operation Core, funded by NIH/NCI Cancer Center Support Grant P30 CA008748, Cycle for Survival, and the Marie-Josée and Henry R Kravis Center for Molecular

Oncology, Memorial Sloan Kettering Cancer Center. All Single Cell Genomics was performed by the Single Cell Analytics and Innovation Lab, Memorial Sloan Kettering Cancer Center. JAZ and DP are investigators of the Howard Hughes Medical Institute.

## Additional information

### Competing interests

Dana Pe'er: is on the scientific advisory board of Insitro. The other authors declare that no competing interests exist.

### Funding

| Funder | Grant reference number | Author |
| --- | --- | --- |
| National Institute of Neurological Disorders and Stroke | NS098832 | Eric R Brooks |
| Howard Hughes Medical Institute | | Dana Pe'er Jennifer A Zallen |
| Alan and Sandra Gerry Metastasis and Tumor Ecosystems Center | | Andrew R Moorman Max Land Roshan Sharma |
| National Cancer Institute | P30 CA008748 | Dana Pe'er Jennifer A Zallen |
| North Carolina State University | | Eric R Brooks |

The funders had no role in study design, data collection and interpretation, or the decision to submit the work for publication.

### Author contributions

Eric R Brooks, Conceptualization, Software, Formal analysis, Funding acquisition, Investigation, Visualization, Methodology, Writing – original draft, Writing – review and editing; Andrew R Moorman, Data curation, Software, Formal analysis, Visualization, Methodology, Writing – review and editing; Bhaswati Bhattacharya, Ian S Prudhomme, Investigation, Methodology; Max Land, Formal analysis; Heather L Alcorn, Formal analysis, Investigation, Methodology; Roshan Sharma, Supervision, Methodology, Writing – review and editing; Dana Pe'er, Supervision, Funding acquisition, Methodology; Jennifer A Zallen, Conceptualization, Formal analysis, Supervision, Funding acquisition, Visualization, Writing – review and editing

### Author ORCIDs

Eric R Brooks ⓘD https://orcid.org/0000-0003-3159-8626
Jennifer A Zallen ⓘD https://orcid.org/0000-0003-3975-1568

### Ethics

All animal experiments were conducted in accordance with PHS guidelines and the NIH Guide for the Care and Use of Laboratory Animals and an approved Institutional Animal Care and Use Committee protocol (15-08-013) of Memorial Sloan Kettering Cancer Center.

Reviewer #1 (Public review): https://doi.org/10.7554/eLife.102819.3.sa1
Reviewer #2 (Public review): https://doi.org/10.7554/eLife.102819.3.sa2
Reviewer #3 (Public review): https://doi.org/10.7554/eLife.102819.3.sa3
Author response https://doi.org/10.7554/eLife.102819.3.sa4

# Additional files

## Supplementary files

Supplementary file 1. Cell type and quality control metrics for wild-type and SAG treatment datasets. The total number of cells post-filtering, the number of cells assigned to each cell type, and quality metrics (median number of UMIs, median number of genes, median percent mitochondrial UMIs, median percent ribosomal UMIs, and mean reads/UMI) are shown for each replicate in the wild-type, SAG treatment, and SAG control datasets.

Supplementary file 2. PhenoGraph clustering of all cranial cells in E7.5–9.0 embryos. Gene expression for each PhenoGraph cluster in the full cranial cell dataset. The false-discovery rate (FDR) adjusted p-value, MAST hurdle value, and empirical $\log_2$ fold change are shown for each gene.

Supplementary file 3. PhenoGraph clustering of cranial neural plate cells in E7.5–9.0 embryos. Gene expression for each PhenoGraph cluster in the cranial neural plate are shown. The false-discovery rate (FDR) adjusted p-value, MAST hurdle value, and empirical $\log_2$ fold change are shown for each gene.

Supplementary file 4. Temporally regulated genes in the forebrain, midbrain/r1, and hindbrain of E7.5–9.0 embryos. Gene correlation scores for the temporally correlated diffusion component (DC0) in each region are shown for the forebrain, midbrain/r1, and hindbrain cranial neural plate populations. The cluster number, Pearson correlation coefficient, and normalized range of expression are shown for each gene.

Supplementary file 5. Diffusion component gene correlations in the E8.5–9.0 cranial neural plate. Gene correlations with each of the top 10 diffusion components in the E8.5–9.0 cranial neural plate are shown. The false discovery rate (FDR) adjusted p-value and Pearson correlation coefficient for each diffusion component are shown for each gene.

Supplementary file 6. Spatially patterned genes in the E8.5–9.0 cranial neural plate. All genes detected in cells from wild-type E8.5–9.0 embryos are shown. The results of HotSpot analysis are given, including the Geary's C spatial autocorrelation coefficient (C, MSD), the z-scored coefficient (Z, MSD), the p-value (Pval, MSD), the false discovery rate adjusted p-value (FDR, MSD), and the normalized range of expression for each gene, in addition to the results of HotSpot analysis for the anterior-posterior (AP) axis-correlated diffusion component (DC0) and the mediolateral (ML) axis-correlated diffusion component (DC2). Cluster identities along the AP axis (DC0), ML axis (DC2), and at various 2D distance (D) values are shown.

Supplementary file 7. Comparison of predicted gene expression patterns to published data. For genes predicted to be spatially patterned in *Supplementary file 6*, the predicted expression of genes in anterior-posterior (AP) or mediolateral (ML) clusters was compared with published images in the Mouse Genome Informatics Gene Expression Database. In addition to the information in *Supplementary file 6*, the predicted and observed expression patterns and whether they agree is reported and a link to available images of wild-type embryos from Theiler stages 12–14 is provided.

Supplementary file 8. Spatially patterned transcriptional regulators in the E8.5–9.0 cranial neural plate. Spatial (2D) cluster (distance D=10), AP cluster, and ML cluster of transcriptional regulators in the E8.5–9.0 cranial neural plate are shown.

Supplementary file 9. Spatially patterned ligands and receptors in the E8.5–9.0 cranial neural plate. Spatial (2D) cluster (distance D=10), AP cluster, ML cluster, and NCBI GeneID of genes assigned as ligands or receptors by the indicated source databases are shown.

Supplementary file 10. Differentially expressed genes in SAG-treated and control embryos in the E8.5 cranial neural plate. Results of MAST differential gene expression analysis in SAG-treated vs, control embryos separated by forebrain, midbrain/r1, and hindbrain regions. The false discovery rate (FDR) adjusted p-value, MAST hurdle value, and empirical $\log_2$ fold change value are shown for each gene.

Supplementary file 11. Regional analysis of transcriptional changes upon SAG treatment. Genes with significant expression changes from *Supplementary file 10* (FDR-adjusted p-value $P<0.001$) grouped by region, including genes deregulated in all cranial neural plate regions (FB-MB-HB), two regions (FB-MB, FB-HB, MB-HB), or single regions (FB, MB, HB). Upregulated genes with a MAST hurdle value >0.24 are highlighted in green. Downregulated genes with a MAST hurdle value <–0.24 are highlighted in red.

Supplementary file 12. Genes used for cell cycle normalization. Gene ontology (GO) terms associated with the cell cycle and the associated genes used in cell cycle normalization are listed.

Supplementary file 13. Genes used for cell typing. Genes used to assign cell/tissue type are listed, along with their positive or negative expression in that tissue and any expression overlap in other tissues.

MDAR checklist

## Data availability

All single-cell datasets generated in this study are publicly available in the Gene Expression Omnibus, accession number GSE273804. Processed h5ad files to examine gene expression patterns in cellxgene (https://github.com/chanzuckerberg/cellxgene; *Megill et al., 2024*) or scanpy (https://github.com/scverse/scanpy; *Wolf et al., 2025*; *Wolf et al., 2018*) are also available in the Gene Expression Omnibus as supplementary files under the same accession number. Code used to analyze gene expression and generate the figures in this study, as well as code used to automatically generate customizable links to gene expression images in the Mouse Genome Informatics Gene Expression Database, are available on GitHub (https://github.com/ZallenLab/Brooks-et-al-scRNAseq-analysis-notebook, copy archived at *Brooks, 2025a*; https://github.com/ZallenLab/MGI-Gene-eXpression-Database-search, copy archived at *Brooks, 2025b*).

The following dataset was generated:

| Author(s) | Year | Dataset title | Dataset URL | Database and Identifier |
|---|---|---|---|---|
| Brooks E, Moorman A, Bhattacharya B, Prudhomme I, Land M, Alcorn H, Sharma R, Pe'er D, Zallen J | 2024 | A single-cell atlas of spatial and temporal gene expression in the mouse cranial neural plate | https://www.ncbi.nlm.nih.gov/geo/query/acc.cgi?acc=GSE273804 | NCBI Gene Expression Omnibus, GSE273804 |

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
