## [Editor Report · eLife Assessment]

This comprehensive scRNAseq atlas of the cranial region during neural induction, patterning, and morphogenesis provides a **fundamental** demonstration of how different cell fates are organized in specific spatial patterns along the anterior-posterior and medial-lateral axes within the developing neural tissue. The **compelling** data are analyzed with a rigorous computational approach, and the data revealed both known and novel genes differentially expressed along rostro-caudal and medio-lateral axes. This will be a helpful resource for researchers studying brain development.

---

## [Referee Report · Reviewer #1 (Public review)]

Summary:

This impressive study presents a comprehensive scRNAseq atlas of the cranial region during neural induction, patterning, and morphogenesis. The authors collected a robust scRNAseq dataset covering six distinct developmental stages. The analysis focused on the neural tissue, resulting in a highly detailed temporal map of neural plate development. The findings demonstrate how different cell fates are organized in specific spatial patterns along the anterior-posterior and medial-lateral axes within the developing neural tissue. Additionally, the research utilized high-density single-cell RNA sequencing (scRNAseq) to reveal intricate spatial and temporal patterns independent of traditional spatial techniques.

The investigation utilized diffusion component analysis to spatially order cells based on their positioning along the anterior-posterior axis, corresponding to the forebrain, midbrain, hindbrain, and medial-lateral axis. By cross-referencing with MGI expression data, the identification of cell types was validated, affirming the expression patterns of numerous known genes and implicating others as differentially expressed along these axes. These findings significantly advance our understanding of the spatially regulated genes in neural tissues during early developmental stages. The emphasis on transcription factors, cell surface, and secreted proteins provides valuable insights into the intricate gene regulatory networks underpinning neural tissue patterning. Analysis of a second scRNAseq dataset where Shh signaling was inhibited by culturing embryos in SAG identified known and previously unknown transcripts regulated by Shh, including the Wnt pathway.

The data includes the neural plate and captures all major cell types in the head, including the mesoderm, endoderm, non-neural ectoderm, neural crest, notochord, and blood. With further analyses, this high-quality data promises to significantly advance our understanding of how these tissues develop in conjunction with the neural tissue, paving the way for future breakthroughs in developmental biology and genomics.

Strengths:

The data is well presented in the figures and thoroughly described in the text. The quality of the scRNAseq data and bioinformatic analysis is exceptional.

Weaknesses:

None

---

## [Referee Report · Reviewer #2 (Public review)]

Summary:

Brooks et al. generate a compelling gene expression atlas of the early embryonic cranial neural plate. They generate single-cell transcriptome data from early cranial neural plate cells at 6 consecutive stages between E7.5 to E9. Utilizing computational analysis they infer temporal gene expression dynamics and spatial gene expression patterns along the anterior-posterior and mediolateral axis of the neural plate. Subsequent comparison with known gene expression patterns revealed a good agreement with their inferred patterns, thus validating their approach. They then focus on Sonic Hedgehog (Shh) signalling, a key morphogen signal, whose activities partition the neural plate into distinct gene expression domains along the mediolateral axis. Single-cell transcriptome analysis of embryos in which the Shh pathway was pharmacologically activated throughout the neural plate revealed characteristic changes in gene expression along the mediolateral axis and the induction of distinct Shh regulated gene expression programs in the developing fore-, mid- and hindbrain.

Strengths:

This manuscript provides a comprehensive transcriptomic characterisation of the developing cranial neural plate, a part of the embryo that to my knowledge has not been extensively analysed by single-cell transcriptomic approaches. The single-cell sequencing data appears to be of high quality and will be a great resource for the wider scientific community. Moreover, the computational analysis is well executed and the validation of the sequencing data using published gene expression patterns is convincing. In my opinion the authors completely achieved their aim of generating a reliable sequencing atlas of the early cranial neural plate. Conceptually, the findings that gene expression patterns differ along the rostrocaudal, mediolateral and temporal axes of the neural plate and that Shh signalling induces distinct target genes along the anterior-posterior axis of the nervous system are not completely unexpected. However, the comprehensive characterization of the spatiotemporal gene expression patterns and how they change upon ectopic activation of the Shh pathway will definitely contribute to a better understanding of neural plate patterning. Taken together, this is a well-executed study that describes a relevant scientific resource that will likely be of great use for the wider scientific community .

Weaknesses:

No weaknesses were identified.

---

## [Referee Report · Reviewer #3 (Public review)]

Summary:

The authors performed a detailed single-cell analysis of the early embryonic cranial neural plate with unprecedented temporal resolution between embryonic days 7.5 and 8.75. They employed diffusion analysis to identify genes that correspond to different temporal and spatial locations within the embryo. Finally, they also examined the global response of cranial tissue to a Smoothened agonist.

Strengths:

Overall, this is an impressive resource, well-validated against sets of genes with known temporal and spatial patterns of expression. It will be of great value to investigators examining early stages of neural plate patterning, neural progenitor diversity, and the roles of signaling molecules and gene regulatory networks controlling regionalization and diversification of the neural plate.

Weaknesses:

The manuscript should be considered a resource. Experimental manipulation is limited to analysis of neural plate cells that were cultured in vitro for 12 hours with SAG. They have identified a significant set of previously unreported genes that are differentially expressed in the cranial neural plate. Some additional analyses might help to highlight novel hypotheses arising from this remarkable resource.

Comments on revisions: I am satisfied with the responses of the authors and do not have any further concerns.

---

## [Author Response]

The following is the authors’ response to the original reviews.

**Reviewing Editor Comment:**
Please note that all three reviewers suggested this manuscript would best fit as a resource paper at eLife.
**Reviewer #1 (Public review):**
Summary:This impressive study presents a comprehensive scRNAseq atlas of the cranial region during neural induction, patterning, and morphogenesis. The authors collected a robust scRNAseq dataset covering six distinct developmental stages. The analysis focused on the neural tissue, resulting in a highly detailed temporal map of neural plate development. The findings demonstrate how different cell fates are organized in specific spatial patterns along the anterior-posterior and medial-lateral axes within the developing neural tissue. Additionally, the research utilized high-density single-cell RNA sequencing (scRNAseq) to reveal intricate spatial and temporal patterns independent of traditional spatial techniques.The investigation utilized diffusion component analysis to spatially order cells based on their positioning along the anterior-posterior axis, corresponding to the forebrain, midbrain, hindbrain, and medial-lateral axis. By cross-referencing with MGI expression data, the identification of cell types was validated, affirming the expression patterns of numerous known genes and implicating others as differentially expressed along these axes. These findings significantly advance our understanding of the spatially regulated genes in neural tissues during early developmental stages. The emphasis on transcription factors, cell surface, and secreted proteins provides valuable insights into the intricate gene regulatory networks underpinning neural tissue patterning. Analysis of a second scRNAseq dataset where Shh signaling was inhibited by culturing embryos in SAG identified known and previously unknown transcripts regulated by Shh, including the Wnt pathway.The data includes the neural plate and captures all major cell types in the head, including the mesoderm, endoderm, non-neural ectoderm, neural crest, notochord, and blood. With further analyses, this high-quality data promises to significantly advance our understanding of how these tissues develop in conjunction with the neural tissue, paving the way for future breakthroughs in developmental biology and genomics.Strengths:The data is well presented in the figures and thoroughly described in the text. The quality of the scRNAseq data and bioinformatic analysis is exceptional.Weaknesses:No weaknesses were identified by this reviewer.
**Reviewer #2 (Public review):**
Summary:Brooks et al. generate a gene expression atlas of the early embryonic cranial neural plate. They generate single-cell transcriptome data from early cranial neural plate cells at 6 consecutive stages between E7.5 to E9. Utilizing computational analysis they infer temporal gene expression dynamics and spatial gene expression patterns along the anterior-posterior and mediolateral axis of the neural plate. Subsequent comparison with known gene expression patterns revealed a good agreement with their inferred patterns, thus validating their approach. They then focus on Sonic Hedgehog (Shh) signalling, a key morphogen signal, whose activities partition the neural plate into distinct gene expression domains along the mediolateral axis. Single-cell transcriptome analysis of embryos in which the Shh pathway was pharmacologically activated throughout the neural plate revealed characteristic changes in gene expression along the mediolateral axis and the induction of distinct Shh-regulated gene expression programs in the developing fore-, mid-, and hindbrain.Strengths:This manuscript provides a comprehensive transcriptomic characterisation of the developing cranial neural plate, a part of the embryo that to my knowledge has not been extensively analysed by single-cell transcriptomic approaches. The single-cell sequencing data appears to be of high quality and will be a great resource for the wider scientific community. Moreover, the computational analysis is well executed and the validation of the sequencing data using published gene expression patterns is convincing. Taken together, this is a well-executed study that describes a relevant scientific resource for the wider scientific community.Weaknesses:Conceptually, the findings that gene expression patterns differ along the rostrocaudal, mediolateral, and temporal axes of the neural plate and that Shh signalling induces distinct target genes along the anterior-posterior axis of the nervous system are more expected than surprising. However, the strength of this manuscript is again the comprehensive characterization of the spatiotemporal gene expression patterns and how they change upon ectopic activation of the Shh pathway.
**Reviewer #3 (Public review):**
Summary:The authors performed a detailed single-cell analysis of the early embryonic cranial neural plate with unprecedented temporal resolution between embryonic days 7.5 and 8.75. They employed diffusion analysis to identify genes that correspond to different temporal and spatial locations within the embryo. Finally, they also examined the global response of cranial tissue to a Smoothened agonist.Strengths:Overall, this is an impressive resource, well-validated against sets of genes with known temporal and spatial patterns of expression. It will be of great value to investigators examining the early stages of neural plate patterning, neural progenitor diversity, and the roles of signaling molecules and gene regulatory networks controlling the regionalization and diversification of the neural plate.Weaknesses:The manuscript should be considered a resource. Experimental manipulation is limited to the analysis of neural plate cells that were cultured in vitro for 12 hours with SAG. Besides the identification of a significant set of previously unreported genes that are differentially expressed in the cranial neural plate, there is little new biological insight emerging from this study. Some additional analyses might help to highlight novel hypotheses arising from this remarkable resource.

We thank all three reviewers for their thoughtful and constructive public reviews and believe they nicely capture the contributions of our study. We agree that this article represents a valuable resource for the community and agree with its designation as a Tools and Resources article.

We also thank the reviewers for their useful suggestions for improving the manuscript. In addition to addressing most of their comments, described below, we note that we have changed midbrain-hindbrain boundary (MHB) to rhombomere 1 (r1) throughout the paper and in Tables S4, S7, S10, and S11, as this designation is more closely aligned with the literature on this region. In addition, we added the anterior-posterior and mediolateral cluster identities from our wild-type analysis for the genes that were differentially expressed in SAG-treated embryos in Table S11. Lastly, we have added a new figure (Figure 5—figure supplement 2), as suggested by Reviewer 2, in which we compare our results with the published expression of genes in neural progenitor domains along the dorsal-ventral axis of the spinal cord.

**Reviewer #1 (Recommendations for the authors):**
I have a few small suggestions for improving the presentation of the data.(1) It would be helpful to show illustrations and embryo images of all the stages utilized in the analysis in Figures 1A and B.(2) It was difficult to distinguish all the different colors in Figures 3B and 4B. Could you label, as in Figure 4, supplements 1D, F?(3) I was confused by the position of the color code key for Figure 7D-J, thinking it belonged to panels B and C. Could you put it under the figure/heatmap key so that it is clearly linked to panels D-J?

Thank you for these suggestions. We have incorporated the third suggestion to improve readability, but were not able to make the first two changes due to space limitations.

**Reviewer #2 (Recommendations for the authors):**
I only have a couple of minor additional suggestions/questions for the authors:(1) The authors state that nearly half of the transcripts they found as differentially regulated in SAG-treated embryos were also characterized as spatially regulated in the wild-type embryos. It would be great if the authors could provide more detail here. How many of the transcripts that are differentially regulated along the mediolateral axis of the wild-type are characterized as differentially regulated in the SAG-treated embryos? How does this further break down into where these genes are expressed along the mediolateral and the anterior-posterior axes? I am aware that the authors answer some of these questions already by providing examples, but a more systematic characterisation would be appreciated here.

We have updated Table S11 to include the anterior-posterior and mediolateral cluster identities of differentially expressed genes in SAG-treated embryos, where applicable. In addition, we have added more discussion of the genes from our SAG analysis that were also found to be spatially patterned in wild-type embryos to the fourth paragraph of the last results section.

(2) Related to the previous question, the authors nicely demonstrate that SAG treatment of embryos causes many transcriptional changes, including the expression/repression of several transcription factors well-known to mediate spatial patterning, raising the question of which of these effects are directly due to gene regulation by the Shh pathway and which effects are secondary consequences of transcriptional changes of other transcription factors. Similarly, the authors' results also suggest that some genes are only induced in specific parts along the neuraxis, raising the question of why. The authors could attempt some type of regulon-interference approaches to identify further candidates that may mediate these effects.

This is an excellent suggestion for a future extension of this work, as we agree that validation of the predicted SHH targets, including which targets are direct, indirect, or region-specific, would be required to evaluate the predictions of this scRNA-seq analysis.

(3) The authors report that they observed 'a previously unreported inhibition of Scube2' upon SAG treatment of the embryos. At least in the spinal cord Scube2 is well-known to be expressed at a distance from the source of Shh secretion (e.g. Kawakami et al. Curr. Biol. 2005), thus the direct or indirect repression by Shh signalling is strongly expected. Moreover, a recent preprint (Collins et al. bioRxiv, https://doi.org/10.1101/469239) suggests that the interaction between Shh and Scube2 can mediate the scale-invariance of Shh patterning. Of note, the authors of this preprint also state that 'upregulation of Shh represses scube2 expression while Shh downregulation increases scube2 expression thus establishing a negative feedback loop.'

Thank you for this suggestion. We have added these references.

(4) The authors partition genes based on different diffusion components as being differentially expressed along the mediolateral axis. However, starting from ~e8.5, neural progenitors in the neural tube can be partitioned based on the expression of well-characterised combinatorial sets of transcription factors into molecularly defined progenitor domains that subsequently give rise to functionally distinct types of neurons. How much of this patterning process can the authors capture with their diffusion component analysis and does their data also allow them to capture these finer-grained differences in gene expression along the mediolateral and prospective dorsal-ventral axis of the neural tube that are known to exist?

This is a very interesting point. We have added a new figure showing UMAPs of the E8.5-9.0 cranial neural plate for a subset of 29 genes (described in Delile et al., 2019) that define distinct neural progenitor domains along the dorsal-ventral axis of the spinal cord (Figure 5—figure supplement 2). We observed that 18 of 20 genes that were detected in the midbrain/r1 region in our dataset were expressed in broad domains along the mediolateral axis of the cranial neural plate that were roughly consistent with their expression domains along the dorsal-ventral axis of the spinal cord. Of these 18 genes, 14 were patterned along both anterior-posterior and mediolateral axes, 2 were patterned only along the mediolateral axis, and 2 were patterned only along the anterior-posterior axis. These results suggest a general correspondence between mediolateral patterning in the cranial neural plate and dorsal-ventral patterning in the spinal cord. However, less refinement of these domains along the mediolateral axis was observed in the cranial neural plate, possibly because the relatively early, pre-closure stages captured by our dataset may be before the establishment of secondary feedback systems that lead to fine-scale patterning of mutually exclusive neural precursor domains. These results are described in the last paragraph of the results section titled “An integrated framework for analyzing cell identity in multiscale space.”

(5) The authors state that they will not only make the raw sequencing data but also the processed intermediate data files available. This is greatly appreciated as it strongly facilitates the re-use of the data. However, it would be also appreciated if the authors made the computational code publicly available that was used to analyze the data and generate the figure panels in the manuscript.

We have deposited the processed h5ad files in the GEO database, accession number GSE273804. Additionally, we have made interactive python notebooks available with the code used to analyze gene expression and generate the figures in this study, as well as code used to automatically generate customizable links to gene expression images in the Mouse Genome Informatics Gene Expression database, on our lab GitHub page (https://github.com/ZallenLab). We have updated the Data availability section to reflect these changes.

**Reviewer #3 (Recommendations for the authors):**
(1) Considering that individual progenitor domains in the developing neural tube are typically sharply delineated with few cells exhibiting mixed identities, it is interesting that clustering of single-cell data results in a largely continuous “cloud” of cells. Is this because the early neural plate cells have not yet crystallized their identity, or would clustering based on a smaller set of genes that exhibit high variance across only neural plate cells result in improved granularity, allowing for better characterization and quantification of distinct progenitor subtypes?

Thank you for raising this interesting point. The apparent continuity of gene expression in the cranial neural plate could reflect a gene signature shared by cranial neural plate cells and that cells may not be extensively regionalized into unique populations at these early stages. We now discuss these possibilities in the third paragraph of the discussion.

(2) Can the authors clarify how neural plate cells were identified and how they were distinguished from the anterior epiblast?

Cell typing was performed by supervised clustering based on known markers of fate. Cranial neural plate cells were identified by their expression of pan-neural factors (*Sox2* and *Sox3*), early or late neural plate markers (*Cdh1* or *Cdh2*), and the lack of markers associated with non-neural ectodermal cell fates (*Grhl2, Krt18*, *Tfap2a*) or other cell types (*Ets1, T, Tbx6*). Full gene sets used to identify all cell types in our analysis are provided in Supplementary Table 13.

(3) Did the study identify cells with cranial placode identity? Cranial placodes emerge during the same period, and it would be useful to highlight them in Figure 1.

Thank you for highlighting this point. Examination of the early placode markers *Six1* and *Eya1* indicates that cranial placode cells are a subset of the cells in PhenoGraph cluster 17 in our full dataset Figure 1—figure supplement 1. We now mention this along with other cell types of interest in the last paragraph of the discussion.

(4) It could be interesting to provide more information about the novel genes identified as differentially expressed along the AP or mediolateral axes. Do they belong to gene families that were not previously implicated in neural patterning, or do they point to novel biological mechanisms controlling neural patterning?

Diverse gene families are represented by the genes that are patterned along the anterior-posterior and mediolateral axes of the cranial neural plate at these stages, likely due to the large number of genes that are spatially patterned in this tissue. Further investigation of the biological mechanisms suggested by these patterns is an important direction for future work, both in terms of molecularly classifying the genes identified as well as directly investigating their roles in neural patterning using genetic analysis.

(5) It would be helpful to discuss how the data presented here compare to other relevant single-cell analyses, such as PMC10901739. This would help to highlight aspects that are unique to this study.

We have added this reference as well as an earlier study from these authors and we discuss how our study complements this work in the introduction.

(6) The inclusion of single-cell data from control embryos that were cultured for 12 hours is of great interest. The authors should identify the set of genes that are deregulated in cultured cells and, taking advantage of their detailed temporal series, examine whether the maturation of cultured embryos progresses normally or whether there are genes that fail to mature correctly in vitro.

We agree that an analysis of the impact of ex vivo culture on gene expression would be useful. However, the large difference in the number of cells in our wild-type and cultured embryo datasets, as well as the lack of time-course data for the cultured embryos, could make a comparison between our current cultured and non-cultured embryo datasets difficult to interpret.